# Clarifying the Main Root Distribution of Trees in Varied Slope Environments Using Non-Destructive Root Detection

Mochammad Taufiqurrachman [1], Utami Dyah Syafitri [2], Mohamad Miftah Rahman [1,3,*], Iskandar Z. Siregar [4] and Lina Karlinasari [1,*]

1 Department of Forest Products, Faculty of Forestry and Environment, IPB University, Jl. Lingkar Akademik, Dramaga, Bogor 16680, West Java, Indonesia; 101200s2mochammad@apps.ipb.ac.id

2 Department of Statistics, Faculty of Mathematics and Natural Sciences, IPB University, Jl. Lingkar Akademik, Dramaga, Bogor 16680, West Java, Indonesia; utamids@apps.ipb.ac.id

3 Department of Sustainable Bioproducts, Mississippi State University, 201 Locksley Way, Starkville, MS 39759, USA

4 Department of Silviculture, Faculty of Forestry and Environment, IPB University, Jl. Lingkar Akademik, Dramaga, Bogor 16680, West Java, Indonesia; siregar@apps.ipb.ac.id

* Correspondence: miftahrahman@apps.ipb.ac.id (M.M.R.); karlinasari@apps.ipb.ac.id (L.K.)

**Abstract:** Tree stability relies on the characteristics of both root and crown structures. However, studying root systems is challenging due to their underground location, often requiring destructive methods for assessment. Non-destructive approaches offer potential solutions, such as the root detector tool. However, research in this area remains limited and requires further development. This study aims to evaluate the root detector tool by inspecting the radial root distribution in trees with different tree crown shapes, both excurrent (*Agathis loranthifolia*) and decurrent (*Samanea saman*), which grow in various soil slopes and soil slope positions. In addition, we establish correlations between tree morphometry, the physical properties of soil, root attributes, sound wave velocity, and their relationship. Based on the results, it was found that the root detector tool is effective in evaluating root distribution, including identifying the main root. The slope position of the tree in a slope class influences the radial distribution of the main roots. This is related to the crown growth as indicated by the direction of its crown. Principal Component Analysis (PCA) findings suggest that parameter morphometric and soil and root properties data clustering align with slope position rather than slope class.

**Keywords:** tree stability; tree morphometric; tree crown; root detector; sonic velocity; slope class

## 1. Introduction

Root architecture refers to the horizontal and vertical distribution of roots, which significantly affects the stability of a tree [1]. This stability is influenced by various factors such as the tree's nutrition, its ability to distribute roots in the soil, the mechanical strength of its roots [2], the physical and mechanical properties of the soil [3], the depth and shape of the roots [4], the interaction between the trunk and roots [5], and the number of roots which are important factors for tree static stability [6]. However, rather than just roots, according to Nunes et al. [7], the primary factor affecting the stability of a stand is a combination of biological and physical elements as whole tree conditions, such as tree dimensions (height, diameter crown size, and slenderness coefficient), tree vitality and health, as well as the clustering of trees within the stand (stand density).

The distribution of tree roots, both radially and vertically, plays a critical role in various plant functions, including nutrient and water acquisition and mechanical anchorage. This ultimately determines the stability of the tree [1]. However, as root architecture is not visible, studying it can be challenging but crucial, given its direct impact on tree metabolism and stability. There are numerous techniques to reveal the root system. Root systems differ

between species and, importantly, affect tree stability through root anchorage [8]. Most of the research is carried out using destructive methods, which involve excavating the soil and then undertaking root mapping and measuring the biomass of the roots of a tree. However, these methods can damage the tree. Employing non-destructive methods is advantageous as they do not harm the tree, enabling them to be repeated multiple times on the same tree. The long-term study of the root system and its growth is possible [9]. Tools and techniques have been developed to determine tree roots and avoid excavation that could harm them. Non-destructive testing techniques like ground-penetrating radar (GPR) are used to study root architecture and distribution [10–13] alongside electrical impedance tomography (EIT) [14] and X-ray computed tomography [15]. Despite its usefulness, GPR has limitations in distinguishing roots from utility lines, restricting its use to urban areas [16], and it also is ineffective in soils with high clay and water content [15]. In addition, these techniques are still costly; thus, there is no efficient and inexpensive non-destructive testing tool that can map the root system or detect the presence of large individual roots [17]. The development of electrical impedance tomography (EIT) for root evaluation has been ongoing since the 1970s [14]. EIT is already used for measuring root length and root mass [14,18–20]. A root detector is another tool that is simpler to detect the presence of roots as a non-destructive method based on the principle of the sonic propagation of material by utilizing different time-of-flight waves as they pass through roots and soil. Research has been conducted on these principles and methods for detecting coarse roots in sloping soil [21]. However, there is still a need for more research on the impact of sloping and how it affects growth development, which is related to tree stability.

Tree stability is not only related to the root distribution but is also closely linked to the growth and development of crown characteristics. Furthermore, the crown of the trees depends on their environment. The trees that grow close to their neighbor become tall and thin with little side branch development and small crowns. In urban areas, trees often grow as a single tree, develop significant side branches, and have greater trunk thickening, making them more stable as decurrent or excurrent growth types based on the species [22]. To better understand the effects of environmental factors on tree root characteristics, specifically the radial distribution of tree roots in slope conditions concerning tree stability, additional studies on root distribution in tree stability with sloping and crown characteristics as variables are necessary. Consequently, the objectives of this study are to evaluate the non-destructive method using the root detector to inspect the radial root distribution in different tree crown shapes in terms of the growth type, excurrent (*Agathis loranthifolia*), and decurrent (*Samanea saman*) trees which grow in various soil slope and soil slope positions, and to determine the relationship between biological factors such as tree morphometry, the physical properties of roots, and the soil's physical properties.

## 2. Materials and Methods

### 2.1. Sampling and Site Description

This research was carried out on twelve rain trees (*Samanea saman*) (Figure 1a) representing the decurrent tree and twelve damar trees (*Agathis loranthifolia*) (Figure 1b) representing the excurrent tree. Damar trees are located at the Gunung Walat Education Forest (GWEF) owned by IPB University in the Cicantayan district, Sukabumi, West Java (latitude 6°54′23″–6°55′35″ south, longitude 106°48′27″–106°50′29″ east) which has a landscape with varying slopes and elevation, ranging from mountains and hills to a flat landscape. The topography varies from sloping to undulating, especially in the south, while the north has a steeper topography [23].

Meanwhile, rain trees are located at the Landscape of IPB University, Dramaga district, Bogor, West Java (6°32′41″–6°33′58″ LS, and 106°42′47″–106°44′07″ BT). The topography varies from flat to undulated and steep slopes. Many habitat types in the campus area are important flora and fauna habitats, including natural ponds and swamps, rivers, arboretums, plantations, shrubs, and grassland. More than two-thirds of the campus is a green open space [24].

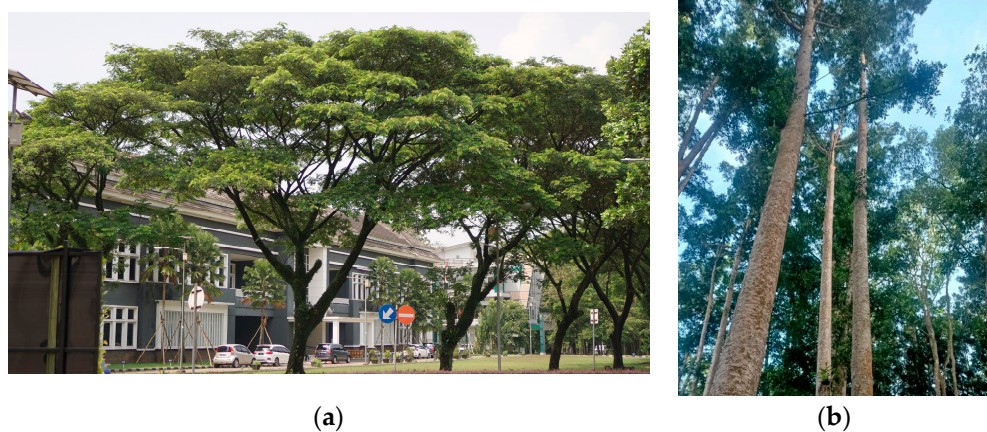

(**a**)　　　　　　　　　　　　　　　　　　(**b**)

**Figure 1.** Tree samples: (**a**) rain trees (*Samanea saman*) as decurrent trees; (**b**) damar trees (*Agathis loranthifolia*) as excurrent trees.

Tree samples were collected for each species within specific slope classes following the CORINE classification model [25]. The slopes were categorized into the following four classes: gentle to flat (class 1, 0–5%), gentle (class 2, 6–15%), steep (class 3, 16–30%), and very steep (class 4, ≥31%). Slope measurements were conducted using a DUKA tool (Digital Protractor Inclinometer Laser Level Portable—LI1). A comprehensive data collection effort was undertaken, encompassing parameters such as slope angles, tree dimensions, tree morphometry, physical soil properties, physical root properties, root detection, and visual root mapping. Observations were made based on the slope position of the sample trees, with a distinction drawn between up-slope (US) and down-slope (DS). Both tree species had a diameter range of approximately 40–59 cm and an average total height of more than 15 m.

### 2.2. Tree Morphometric

Tree morphometrics encompasses a tree's size and physical attributes, including measurements such as trunk diameter, tree height, and crown shape. Tree dimensions and crown characteristics were assessed using a phi band, meter tape, and haga hypsometer. According to Anselmo et al. [26], there is a positive correlation between the root and crown areas. Therefore, this study adopted two tree morphometric parameters, the live crown ratio (*LCR*) and mean crown diameter (*DCR*), to investigate their correlation with root distribution, following the approach described by Karlinasari et al. [27] and Rahman et al. [28].

The *LCR* was calculated by dividing the crown length by the tree height ($LCR = h_{cr}/h$). The measurement used crown projection in eight subcardinal directions. It was then determined by the longest and shortest crown length directions from the tree. This method was modified by Pretzsch et al. [29]. Meanwhile, *DCR* was calculated as the average of the longest and shortest crown diameters.

### 2.3. Physical Properties of Soil

Soil physical properties, including bulk density (*BD*), porosity (*Po*), soil moisture content (*MCs*), and soil relative humidity (*RH*), were measured by collecting undisturbed soil samples using a ring sample with dimensions of 4 cm in height and ±7.6 cm in diameter. The soil moisture measurement was performed according to Wijayanto et al. [30], where three repetitions were carried out at 10 min intervals in both the up-slope and down-slope positions at a depth of 20 cm. The calculation of the soil's physical properties was modified, as shown in Equations (1)–(3). Here, *Wfs* = the weight of fresh soil + ring (g); *Wds* = the

weight of oven-dried soil (g); $Wr$ = the weight of the ring (g); $Vs$ = the volume of the soil sample (cm$^3$); 2.65 = the soil particle density (g·cm$^{-3}$).

$$BD\left(\text{g·cm}^{-3}\right) = \frac{Wds}{Vs} \tag{1}$$

$$Po(\%) = \left(1 - \frac{BD}{2.65}\right) \times 100\% \tag{2}$$

$$MCs(\%) = \frac{(Wfs - Wr) - Wds}{Wds} \times 100\% \tag{3}$$

*2.4. Physical Properties of Root Mass*

For each tree sample, woody root mass samples with dimensions of 2 × 2 × 1 cm were collected from the up-slope and down-slope root positions for each tree sample. Afterward, the physical properties of the root moisture content (*MCr*) and woody root biomass density (ρ) were determined. The fresh weight of the root samples (*Wfr*) was measured, followed by the root sample volume (*Vr*), which was determined by Archimedes' principal method. Subsequently, the root samples underwent oven-drying at 105 °C for 24 h until a constant weight was achieved to obtain the dry weight (*Wdr*). The physical properties of the roots were then calculated based on Equations (4) and (5).

$$\rho\left(\text{g·cm}^{-3}\right) = \frac{Wfr}{Vr} \tag{4}$$

$$MCr\ (\%) = \frac{Wfr - Wdr}{Wdr} \times 100\% \tag{5}$$

*2.5. Root Detection Measurement*

The estimation of root radial distribution was performed using the Fakopp® root detector (Fakopp Enterprise Bt., Agfalva, Hungary) tool that relies on the time-of-flight (ToF) of sound propagation. The detection of roots is accomplished by measuring the speed of the acoustic signal as it travels through the soil and the woody root biomass. Variations in the density of these materials result in differences in the acoustic signal, allowing for the identification of the presence or absence of the main roots. This tool consisted of a receiver sensor, a transmitter sensor, a dual amplifier box, and a battery box, with sensors equipped with spikes to facilitate penetration into the tree stem and soil (Figure 2b). The transmitter sensor was inserted into the tree trunk at the root neck position and oriented at a 45° angle to the trunk, while the receiver sensor was inserted at a 45° angle to the ground surface. The distance between the receiver sensor and the center of the stem was maintained at 80 cm. Based on research by Proto et al. [1], this 80 cm distance was chosen due to its correlation between the speed of sound waves and the density of woody root biomass compared to the distances of 40 and 120 cm. To identify the presence of roots in the soil, the root detector tool utilized a transmitter sensor that generated a sound wave upon being struck by a hammer. This wave traveled and was subsequently detected by a receiver sensor, with the time elapsed between generation and detection being measured and recorded. The root detector tool accurately determined the root location by analyzing the timing of wave propagation.

Acoustic signal measurements were conducted in an orbital pattern around the stem, with measurement points situated at intervals of 10–15 cm (approximately 11°) apart, commencing from the north side and proceeding clockwise (Figure 2a) [9]. Sound wave velocity data were collected by hitting the transmitter sensor with a hammer to initiate sound propagation, which was subsequently captured by the receiving sensor. The hammer-hitting procedure was repeated three times until a consistent sound wave propagation time was obtained. The data for the time of flight over a set distance was recorded and then

analyzed using the root detector evaluation 2.4 application (Fakopp Enterprise Bt, Agfalva, Hungary) to provide sonic velocity data.

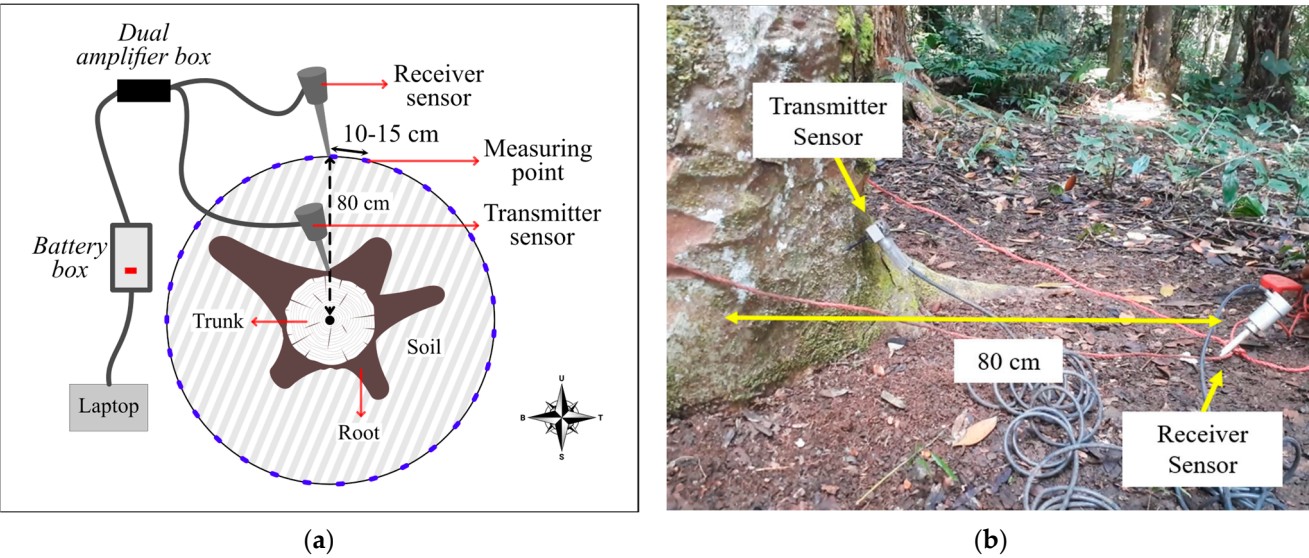

(**a**)  (**b**)

**Figure 2.** An experimental setup in the field for root detector testing: (**a**) Layout of the installation setup; (**b**) Fakopp root detector set in the tree.

*2.6. Root Architecture Analysis*

The root detector tool was used to assess the radial distribution of roots, with a particular emphasis on shallow and coarse roots [21]. The process of visualizing the radial root distribution using Microsoft Excel 2019 (Microsoft, Redmond, WA, USA) and Root Detector Evaluation Software 2.4 (Fakopp Enterprise, Agfalva, Hungary) (Figure 3a) was employed. The value of sound wave velocity generated at each point is denoted as "$V$". However, in a large tree with various root sizes, root distribution was commonly determined by the main structural roots. It was possible that 2–3 observation points detected a main root due to the size of these roots [28]. Consequently, further analysis is required to differentiate the main root. Sound wave velocity at the main root was estimated based on the following criteria: a value above the overall average, above 400 m·s$^{-1}$, and a peak value. The "$V_{root}$" denotes the sound wave velocity value that meets these criteria (Figure 3c). The velocity of the 400 m·s$^{-1}$ value pointed out the average velocity of the soil material.

In addition to root detection tools, this research also mapped the root distribution visually using the photogrammetry method (Figure 3b). Photogrammetry is a scientific and artistic approach used to acquire precise measurement and visual data in a three-dimensional (3D) form using two or more captured images. The principle is capturing photos (2D) in an overlapping manner in stereoscopic, then processing them into rendered 3D shapes accurately over a wide range of scales [31]. Photogrammetry methods have previously been used to measure root volume [32] and root distribution [21]. In the photogrammetry method, a smartphone camera with a resolution of 13 MP (f1.9) was used to generate 2D photos, which were then processed using structure-from-motion (SfM) and a Multi-View Stereo (MVS) pipeline. Photos were systematically taken in an orbital pattern, with a minimum of 30 photos captured at distances ranging from 2 to 4 m from the sample tree. Furthermore, photos were further analyzed using the open-source COLMAP (CNR, Palermo, Italy) and Meshlab 2020.07 (ETH, Zurich, Switzerland, and UNC, Chapel Hill, NC, USA) to generate 3D representations [21]. This visual mapping approach provided information about the direction of the roots above the ground, the soil's visual slope, and the base of the tree's stem. Such information proved important for validating the results obtained from root detection tools regarding the direction and distribution of roots [21].

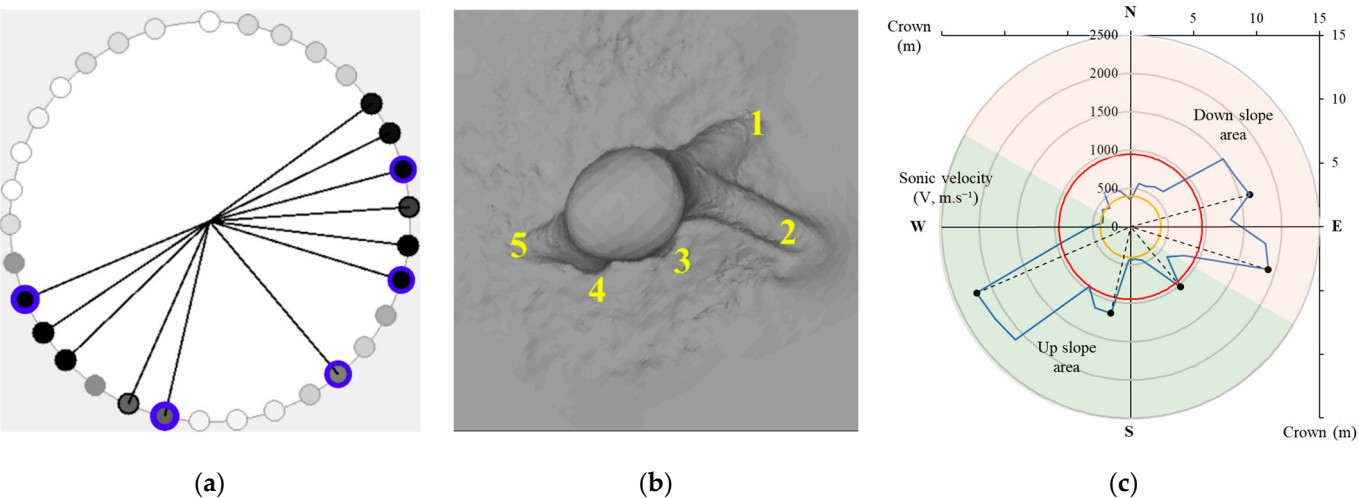

(a)　　　　　　　　(b)　　　　　　　　(c)

**Figure 3.** An example of representative graphical visualization of root distribution based on (**a**) Root Detector Evaluation Software where dark and blue circles indicate the higher sonic velocity values of the main root; (**b**) Visual mapping of root distribution using the photogrammetry method (yellow numbers indicating the number visually detected main root); (**c**) Sonic velocity and distribution data processing using Microsoft Excel (sonic velocity in main root noted as "$V_{root}$" and shown as a blue line with the peak in a dark point; sonic velocity generally noted as "$V$" and shown by a blue line; average of $V$ shown in red circle; $V_{root}$ threshold (400 m·s$^{-1}$) in yellow circle).

For a better understanding of the correlation between the root distribution, slope, and tree canopy type, it is crucial to obtain data on canopy distribution. These data can be collected using photogrammetry and root detectors to visualize the radial root distribution and soil slope direction. After collecting these data, it can be overlaid on a graph that shows the tree canopy projections for each tree, as depicted in Figure 4.

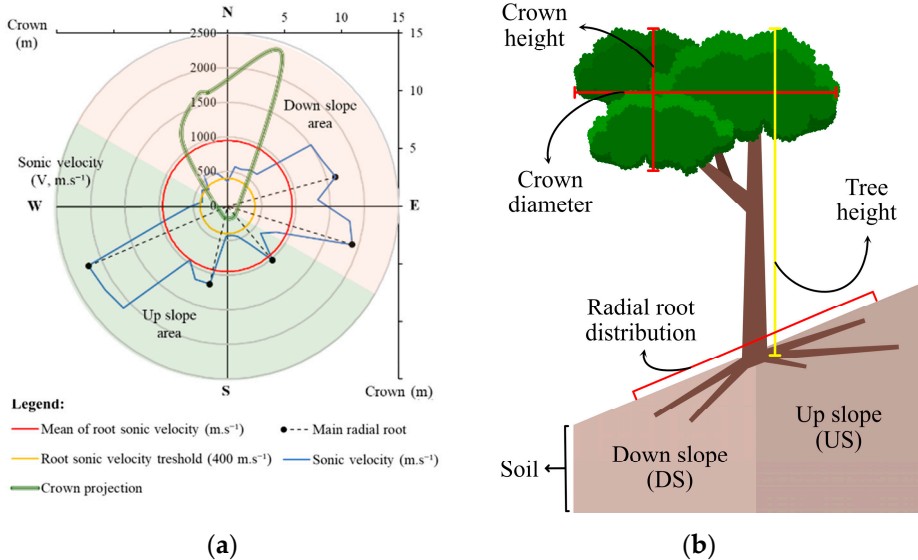

(a)　　　　　　　　(b)

**Figure 4.** Representative image of root detector analysis on slope condition: (**a**) Crown projection with a green line and the distribution of the main radial roots based on the peak of sonic velocity (black dash line) and slope position; (**b**) Crown direction tendency and radial root distribution based on slope position (downslope).

## 2.7. Statistical Analysis

Data processing was carried out using Microsoft Excel 2019 (Microsoft, Redmond, WA, USA). Data were further processed using Minitab 18 software (Minitab, LLC, State College,

PA, USA). Several statistical analyses were used, including the comparison analysis of the *t*-test and nested-analysis of variance (ANOVA) to identify the effect of the slope class and slope position in the soil and root parameters, and Spearman correlation analysis to find a relationship among the used parameters. Since the woody root biomass at different angles measurement in the tree cannot be compared, only one tree was used. The selected tree was located in an area where the slope angle of the land was between the slope class boundaries. The analysis used angular ANOVA to evaluate the impact of soil slope on the sound wave propagation parameter (*V*) using the conversion of sonic data to angular data. Principal Component Analysis (PCA) was also carried out with testing to evaluate the main parameters of this study.

## 3. Results

### 3.1. Tree Morphometric

Table 1 shows the mean growth and morphometric characteristics of the rain trees and damar trees. The diameter and height of rain trees in our study were 49.58 cm and 18.83 m, respectively, while in damar trees, there was an average diameter of 47.25 cm and tree height of 22.42 m. Live crown ratio (*LCR*) and crown diameter (*DCR*) for each slope class are presented in Figure 5. Notably, the average *LCR* value of damar trees is 50.41%, which is higher than that of rain trees' average by 48.28% (Table 1). Except for slope class 4 for damar trees, all trees had *LCR* values of more than 40% (Figure 5a). Meanwhile, for crown diameter (*DCR*), the excurrent growth type of rain trees had more than two times the width of damar trees of the decurrent growth type (Table 1). The reason for using *DCR* in this study was its association with root distribution [27]. Rain trees have more dominant branch growth, resulting in a crown shape that spreads out. Conversely, damar trees are dominated in the apical direction, causing a conical crown shape [33].

**Table 1.** Tree growth and morphometric parameters for rain trees (*Samaea saman*) and damar trees (*Agathis loranthifolia*).

| Parameter | Tree Species | |
|---|---|---|
| | *Samanea saman* (*n* = 12) | *Agathis loranthifolia* (*n* = 12) |
| Diameter (cm) | 49.58 ± 5.91 | 47.25 ± 6.37 |
| Height (m) | 18.83 ± 3.41 | 22.42 ± 1.89 |
| *LCR* (%) | 48.28 ± 13.28 | 50.41 ± 28.38 |
| *DCR* (m) | 14.67 ± 2.35 | 5.96 ± 1.29 |

Note: *LCR*: live crown ratio, *DCR*: mean crown diameter.

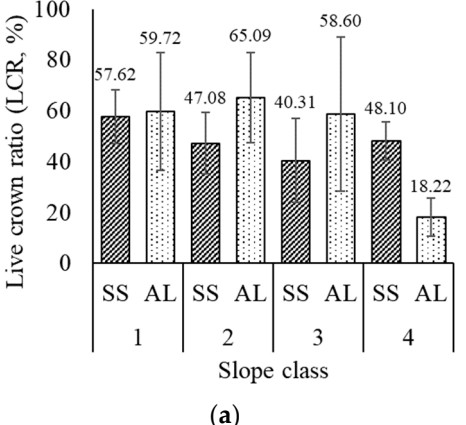
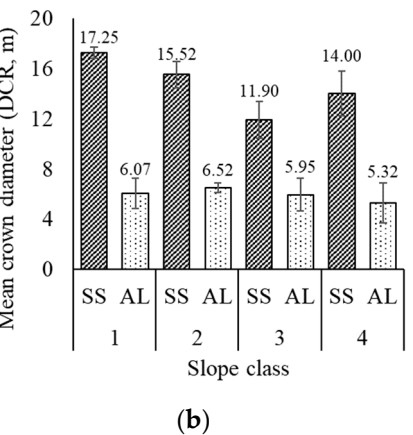

(a)          (b)

**Figure 5.** The average of *LCR* (**a**) and *DCR* (**b**) of damar trees (*n* = 12) and rain trees (*n* = 12) based on the class of the soil slope class (SS = rain trees; AL = damar trees).

### 3.2. Soil Physic Properties

Soil physic properties are related to the site and condition of the environment. The average values of soil physics properties, including bulk density (*BD*), porosity (*Po*), soil moisture content (*MCs*), and soil relative humidity (*RH*) are shown in Table 2.

**Table 2.** Average value of soil physic properties in around of rain trees (*Samanea saman*) and damar trees (*Agathis loranthifolia*) related to slope class and slope position.

| Slope Class | Species | Slope Position | Parameter | | | |
|---|---|---|---|---|---|---|
| | | | $BD$ (g·cm$^{-3}$) | $Po$ (%) | $MCs$ (%) | $RH$ (%) |
| 1<br>(0–5%) | *Samanea saman* (*n* = 3) | NA | 1.05 | 60.44 | 46.57 | 82.00 |
| | *Agathis loranthifolia* (*n* = 3) | NA | 0.77 | 70.85 | 81.29 | 72.50 |
| 2<br>(6–15%) | *Samanea saman* (*n* = 3) | Up-slope<br>Down-slope | 0.93<br>0.95 | 65.08<br>63.98 | 50.34<br>47.02 | 93.33<br>88.33 |
| | *Agathis loranthifolia* (*n* = 3) | Up-slope<br>Down-slope | 0.82<br>0.83 | 69.11<br>68.49 | 77.27<br>68.00 | 96.67<br>100.00 |
| 3<br>(16–30%) | *Samanea saman* (*n* = 3) | Up-slope<br>Down-slope | 1.03<br>0.92 | 61.08<br>65.43 | 43.67<br>48.43 | 100.00<br>96.67 |
| | *Agathis loranthifolia* (*n* = 3) | Up-slope<br>Down-slope | 0.76<br>0.80 | 71.25<br>69.89 | 81.28<br>73.20 | 98.33<br>96.67 |
| 4<br>(≥31%) | *Samanea saman* (*n* = 3) | Up-slope<br>Down-slope | 0.98<br>0.91 | 63.16<br>65.48 | 41.57<br>42.15 | 90.00<br>98.33 |
| | *Agathis loranthifolia* (*n* = 3) | Up-slope<br>Down-slope | 0.74<br>0.76 | 72.25<br>71.37 | 88.77<br>69.99 | 96.67<br>90.00 |

Note: *BD*: bulk density, *Po*: soil porosity, *MCs*: soil moisture content, *RH*: soil relative humidity, NA: not available.

The bulk density of soil (*BD*) indicates the level of soil density. Our study found that the *BD* of soil around rain trees had 0.98 g·cm$^{-3}$, which was significantly higher than that of damar trees by 0.78 g·cm$^{-3}$, as shown in Tables 2 and 3. The denser the soil, the higher the *BD*, leading to a slower water flow rate (infiltration rate) [34]. The soil porosity (*Po*) showed that the average porosity value of the soil around damar trees was significantly higher than the environment around rain trees (Table 3). The up-slope position had a higher porosity than the down-slope in the environment for damar tree species. Meanwhile, rain trees' porosity value was generally lower in the up-slope position, except for soil class 2 (Table 2). This study found that the soil moisture content (*MC*) at the down-slope position was lower than at the up-slope position in the area around these tree species (Table 2). A statistical analysis of the *t*-test found that the area around the damar tree grown had a significantly higher soil moisture content (77.64%) than rain trees (45.79%), as shown in Table 3. Based on the average soil RH value, there was no significant difference between the environment of damar trees and rain trees around where the tree grew (Table 3). The average soil *RH* was 90% in the environment where the two species of trees studied grew.

The statistical analysis of nested ANOVA found no significant difference in four slope classes and two slope positions related to *BD*, *Po*, *MCs*, and *RH* in the area around the rain trees grown. Meanwhile, significant differences were found in the slope classes in the area of damar trees for the physical soil characteristics of *BD* and *Po*.

**Table 3.** *T*-test for soil physical property parameters in area around tree species of rain trees (*Samanea saman*) and damar trees (*Agathis loranthifolia*).

| Parameter | Tree Species | | *p*-Value |
| :---: | :---: | :---: | :---: |
| | *Samanea saman* (*n* = 12) | *Agathis loranthifolia* (*n* = 12) | |
| *BD* (g·cm$^{-3}$) | 0.98 ± 0.08 | 0.78 ± 0.04 | 0.000 * |
| *Po* (%) | 63.14 ± 3.17 | 70.51 ± 1.64 | 0.000 * |
| *MCs* (%) | 45.79 ± 4.90 | 77.64 ± 6.33 | 0.000 * |
| *RH* (%) | 91.38 ± 9.95 | 90.42 ± 14.68 | 0.813 |

Note: *BD*: bulk density, *Po*: soil porosity, *MCs*: soil moisture content, *RH*: soil relative humidity. * Significant at 5% confidence level.

### 3.3. Physical Properties of Woody Root Biomass

The physical properties of the root mass tested consisted of root moisture content and fresh woody root biomass density. Root moisture content (*MCr*) generally varies at the up-slope and down-slope positions for both tree types (Table 4). However, the moisture content of damar trees was higher and significantly different from the moisture content of the root mass of rain trees (Table 5). The fresh woody root biomass density (*ρ*) values for the two types of trees were almost the same, approximately 0.9–1.0 g·cm$^{-3}$) (Table 5), and the *t*-test results did not show any significant differences (Table 6).

**Table 4.** Average value of root physical properties and sonic velocity of rain trees (*Samanea saman*) dan damar trees (*Agathis loranthifolia*) based on slope class and positions on the slope.

| Slope Class | Species | Position | Parameter | | | |
| :---: | :---: | :---: | :---: | :---: | :---: | :---: |
| | | | *MCr* (%) | *ρ* (g·cm$^{-3}$) | *V* (m·s$^{-1}$) | $V_{root}$ (m·s$^{-1}$) |
| 1 (0–5%) | *Samanea saman* (*n* = 3) | NA | 73.96 | 1.08 | 914.15 | 1407.10 |
| | *Agathis loranthifolia* (*n* = 3) | NA | 55.81 | 0.97 | 864.39 | 1687.83 |
| 2 (6–15%) | *Samanea saman* (*n* = 3) | Up-slope | 102.98 | 0.99 | 756.94 | 1603.42 |
| | | Down-slope | 89.72 | 0.96 | 1025.10 | 1683.40 |
| | *Agathis loranthifolia* (*n* = 3) | Up-slope | 132.88 | 0.91 | 852.98 | 1949.91 |
| | | Down-slope | 126.69 | 1.05 | 1187.24 | 1800.50 |
| 3 (16–30%) | *Samanea saman* (*n* = 3) | Up-slope | 97.13 | 0.99 | 792.74 | 1581.93 |
| | | Down-slope | 95.20 | 0.98 | 876.47 | 1352.21 |
| | *Agathis loranthifolia* (*n* = 3) | Up-slope | 109.86 | 0.85 | 815.89 | 1527.34 |
| | | Down-slope | 115.51 | 0.87 | 910.68 | 1535.32 |
| 4 (≥31%) | *Samanea saman* (*n* = 3) | Up-slope | 99.67 | 1.00 | 773.12 | 1394.83 |
| | | Down-slope | 83.91 | 1.00 | 919.67 | 1783.19 |
| | *Agathis loranthifolia* (*n* = 3) | Up-slope | 138.56 | 1.08 | 647.18 | 1798.29 |
| | | Down-slope | 142.48 | 1.02 | 939.20 | 1738.51 |

Note: *MCr*: root moisture content, *ρ*: woody root biomass density, *V*: velocity, $V_{root}$: velocity of main roots.

The results of the nested ANOVA analysis showed that only the woody root biomass moisture content of damar tree species was significantly different owing to differences in the soil slope classes (Table 6). Meanwhile, in the condition of the slope class and position in other slope classes, there were no significant differences between the two types of trees.

**Table 5.** *T*-test for physical root properties and sound wave propagation parameters of rain trees (*Samanea saman*) and damar trees (*Agathis loranthifolia*).

| Parameter | Tree Species | | *p*-Value |
| | *Samanea saman* (*n* = 12) | *Agathis loranthifolia* (*n* = 12) | |
|---|---|---|---|
| *MCr* (%) | 89.57 ± 11.25 | 134.70 ± 21.98 | 0.000 * |
| $\rho$ (g·cm$^{-3}$) | 1.01 ± 0.06 | 0.97 ± 0.10 | 0.162 |
| *V* (m·s$^{-1}$) | 871.54 ± 144.24 | 910.25 ± 117.48 | 0.495 |
| $V_{root}$ (m·s$^{-1}$) | 1526.65 ± 183.30 | 1709.45 ± 264.77 | 0.046 * |

Note: *MCr*: root moisture content, $\rho$: fresh woody root biomass density, *V*: velocity, $V_{root}$: velocity of main roots. * Significant at 5% confidence level.

**Table 6.** Nested ANOVA results of physical root properties in the area around the rain trees (*Samanea saman*) and damar trees (*Agathis loranthifolia*) grown.

| Parameter | Tree Species | Source | DF | Adj SS | Adj MS | F-Value | *p*-Value |
|---|---|---|---|---|---|---|---|
| BD | *Samanea saman* | Slope class | 3 | 0.04479 | 0.014930 | 1.72 | 0.404 |
| | | Slope position | 3 | 0.02682 | 0.008942 | 0.89 | 0.467 |
| | *Agathis loranthifolia* | Slope class | 3 | 0.019921 | 0.006640 | 0.00 | 0.000 * |
| | | Slope position | 3 | 0.003180 | 0.001060 | 0.15 | 0.930 |
| Po | *Samanea saman* | Slope class | 3 | 63.78 | 21.26 | 1.72 | 0.404 |
| | | Slope position | 3 | 38.20 | 12.73 | 0.89 | 0.467 |
| | *Agathis loranthifolia* | Slope class | 3 | 28.367 | 9.456 | 0.00 | 0.000 * |
| | | Slope position | 3 | 4.529 | 1.510 | 0.15 | 0.930 |
| MCs | *Samanea saman* | Slope class | 3 | 146.94 | 48.98 | 3.72 | 0.335 |
| | | Slope position | 3 | 51.02 | 17.01 | 0.53 | 0.671 |
| | *Agathis loranthifolia* | Slope class | 3 | 249.3 | 83.11 | 0.29 | 0.831 |
| | | Slope position | 3 | 756.3 | 252.10 | 2.10 | 0.139 |
| RH | *Samanea saman* | Slope class | 3 | 847.8 | 282.60 | 8.32 | 0.316 |
| | | Slope position | 3 | 158.3 | 52.78 | 0.41 | 0.746 |
| | *Agathis loranthifolia* | Slope class | 3 | 2654.17 | 884.72 | 0.00 | 0.000 * |
| | | Slope position | 3 | 87.50 | 29.17 | 0.17 | 0.918 |
| MCr | *Samanea saman* | Slope class | 3 | 2028.7 | 676.2 | 3.08 | 0.254 |
| | | Slope position | 3 | 641.9 | 214.0 | 1.12 | 0.369 |
| | *Agathis loranthifolia* | Slope class | 3 | 5929.7 | 1976.57 | 0.00 | 0.000 * |
| | | Slope position | 3 | 128.4 | 42.81 | 0.09 | 0.963 |
| $\rho$ | *Samanea laman* | Slope class | 3 | 0.046268 | 0.015423 | 0.00 | 0.000 |
| | | Slope position | 3 | 0.001007 | 0.000336 | 0.08 | 0.968 |
| | *Agathis loranthifolia* | Slope class | 3 | 0.10594 | 0.03531 | 2.88 | 0.288 |
| | | Slope position | 3 | 0.03792 | 0.01264 | 0.90 | 0.463 |

Note: *BD*: bulk density, *Po*: soil porosity, *MCs*: soil moisture content, *RH*: soil relative humidity, *MCr*: root moisture content, $\rho$: woody root biomass density, * Significant at 5% confidence level.

### 3.4. Root Detection

The root detection study was based on the difference in sound wave propagation between the mass roots and the soil. Sound wave propagation on the mass root was faster than the soil, meaning the value mass root was higher than on the soil ground. According to Bucur [34,35], wave propagation on the soil had a sonic velocity value of 250–400 $m \cdot s^{-1}$, depending on the soil and moisture content condition. It was also mentioned that the sonic velocity of the mass roots could reach 2000–4000 $m \cdot s^{-1}$. The "$V$" was the sound wave speed above the limit value of 400 $m \cdot s^{-2}$. Table 5 shows the $V$ values for various slope classes and positions on the slope in a range of 773.12 $m \cdot s^{-1}$ to 1025 $m \cdot s^{-1}$ and 647 $m \cdot s^{-1}$ to 1187.24 $m \cdot s^{-1}$ for rain trees and damar trees, respectively. The average $V$ values obtained were 871.54 $m \cdot s^{-2}$ for rain trees and 910.25 $m \cdot s^{-2}$ for damar trees. There seemed to be no significant difference between these tree species for the $V$ value of the root (Table 5).

The structural main root sound wave propagation value was determined based on the "$V_{root}$" value, which was at the peak of the sound wave speed value. The value of $V_{root}$ rain trees was in the range from 1394.83 to 1783.19 $m \cdot s^{-1}$, while resin trees were in the range 1394.83 to 1949.91 $m \cdot s^{-1}$ (Table 4). Our study found that the average peak of the sound wave propagation of the root mass ($V_{root}$) was lower than 2000 $m \cdot s^{-1}$, i.e., 1526.65 $m \cdot s^{-1}$ and 1709.45 $m \cdot s^{-1}$ for rain trees and damar trees, respectively (Table 4). The statistical analysis of the comparative *t*-test found a significant difference in the mass root sound wave propagation between the rain tree (*Samanea saman*) and the damar tree (*Agathis loranthifolia*).

Since sound wave propagation measurements on the ground using a root detector were carried out using an orbital pattern at each point with intervals of around 11 degrees, the analysis was performed using one-way ANOVA for angular data for sound wave propagation data. The sound wave speed data were transformed into angular data and measured in radians to facilitate this analysis (Table 7). The result found that the soil slope class significantly affected sound wave propagation in rain trees (*Samanea saman*), while in damar trees, it was not significant. Further analysis using Fisher pairwise comparisons of angular data was carried out to identify the significance of sound wave velocity propagation at the soil slope class in rain trees. The gentle soil class slope (6–15%) was found to be significantly different from flat and very steep slopes (Table 8).

**Table 7.** Result of one-way ANOVA of sound wave propagation based on angular data analysis of rain trees (*Samanea saman*) and damar trees (*Agathis loranthifolia*).

| Tree Species | Source | DF | Adj SS | Adj MS | F-Value | *p*-Value |
|---|---|---|---|---|---|---|
| *Samanea saman* | Slope class | 3 | 67,036 | 22,345 | 2.77 | 0.044 * |
| *Agathis loranthifolia* | Slope class | 3 | 18,116 | 6039 | 0.62 | 0.606 |

* Significant at 5% confidence level.

**Table 8.** Fisher pairwise comparisons of angular data of sound wave propagation based on soil slope class of rain trees (*Samanea saman*) and damar trees (*Agathis loranthifolia*).

| Soil Slope Class | *Samanea saman* | | *Agathis loranthifolia* | |
|---|---|---|---|---|
| | Sonic Velocity ($m \cdot s^{-1}$) | Angular Value (radians) | Sonic Velocity ($m \cdot s^{-1}$) | Angular Value (radians) |
| 1 (0–5%)—flat | 934.06 | 177.8 [a] | 889.06 | 169.3 [a] |
| 2 (6–15%)—gentle | 668.41 | 127.3 [b] | 1002.58 | 190.9 [a] |
| 3 (16–30)—steep | 823.23 | 156.7 [ab] | 1053.44 | 200.6 [a] |
| 4 (≥31%)—very steep | 973.60 | 185.4 [a] | 945.57 | 180.0 [a] |

Different letter means significantly different (α = 0.05).

Based on the data analysis root sound wave speed ($V_{root}$) from the root detector tool (Tables 4 and 5), the number of main structural roots from the two tree species that grew in different sites with various soil slope classes could be determined. The research showed

that the average number of structural main roots was five in rain trees, while the damar tree had six trees (Table 9). A further explanation is provided in Section 3.6.

**Table 9.** Number of main structural roots of rain trees (*Samanea saman*) and damar trees (*Agathis loranthifolia*).

| Parameter | Tree Species | | *p*-Value |
| | *Samanea saman* (*n* = 12) | *Agathis loranthifolia* (*n* = 12) | |
|---|---|---|---|
| $\sum_{root}$ root detector | 5.00 ± 2.00 | 6.00 ± 2.00 | 0.538 |
| $\sum_{root}$ photogrammetry | 4.00 ± 1.00 | 3.00 ± 1.00 | 0.198 |

Note: $\sum_{root}$: number of main roots.

*3.5. Relationship of Tree Morphometric, Soil Physic, Woody Root Biomass, and Sonic Wave Propagation*

Statistical relationships among the parameters were developed in this study. The correlation coefficient was interpreted using the conventional approach of Schober and Schwarte [36]. Except for the relationship between the soil physic properties of *BD*, *Po*, and *MCs*, which were significant to each other, the significance of the relationship between parameters in rain trees is different from that in damar trees (Tables 10 and 11). The effect of the growing environment, especially the adaptation of each type of tree growth (excurrent and decurrent types) to the slope position, presumably led to these differences. In rain trees, the soil moisture content (*MCs*) had a significant positive relationship with DCR and the number of roots ($\sum_{root}$). The number of roots ($\sum_{root}$) was also positively correlated with sound wave propagation (*V*). In damar trees, the soil moisture content (*MCr*) had a significant positive correlation with the live crown ratio (*LCR*). For both tree species adapting to the environment, there was a moderate positive relationship between the value of the ground sound wave propagation (*V*) and the root sound wave speed (*V*root). When determining the number of roots ($\sum_{root}$), a positive but weak relationship was found in relation to the *DCR* model of the crown, which indicated the type of tree growth.

**Table 10.** Results of Spearman correlation test for rain trees (*Samanea saman*).

| Parameter | BD | Po | MCs | RH | LCR | DCR | $\sum_{root}$ | MCr | $\rho$ | V | V_root |
|---|---|---|---|---|---|---|---|---|---|---|---|
| BD | 1 | | | | | | | | | | |
| Po | −1.000 * | 1 | | | | | | | | | |
| MCs | −0.525 * | 0.525 * | 1 | | | | | | | | |
| RH | 0.263 | −0.263 | −0.412 * | 1 | | | | | | | |
| LCR | −0.170 | 0.170 | 0.310 | −0.568 * | 1 | | | | | | |
| DCR | −0.121 | 0.121 | 0.676 * | −0.106 | 0.161 | 1 | | | | | |
| $\sum_{root}$ | −0.294 | 0.294 | 0.439 * | −0.310 | 0.144 | 0.322 | 1 | | | | |
| MCr | −0.190 | 0.190 | −0.177 | 0.143 | −0.101 | 0.047 | −0.002 | 1 | | | |
| $\rho$ | 0.323 | −0.323 | 0.013 | −0.361 | 0.245 | −0.203 | 0.013 | −0.386 | 1 | | |
| V | 0.022 | −0.022 | 0.090 | −0.075 | 0.066 | −0.042 | 0.660 * | −0.183 | 0.126 | 1 | |
| V_root | −0.111 | 0.111 | 0.036 | 0.147 | −0.198 | −0.141 | 0.124 | −0.327 | 0.129 | 0.517 * | 1 |

Note: *BD*: bulk density; *Po*: soil porosity; *MCs*: soil moisture content; *RH*: soil relative humidity; *LCR*: live crown ratio; *DCR*: mean crown diameter; $\sum_{root}$: number of main roots; *MCr*: root moisture content; $\rho$: woody root biomass density; *V*: velocity; *V*root: velocity of main roots; * Significant at 5% confidence level.

**Table 11.** Results of Spearman's correlation test of damar trees (*Agathis loranthifolia*).

| Parameter | BD | Po | MCs | RH | LCR | DCR | $\sum_{root}$ | MCr | $\rho$ | V | $V_{root}$ |
|---|---|---|---|---|---|---|---|---|---|---|---|
| BD | 1 | | | | | | | | | | |
| Po | −1.000 * | 1 | | | | | | | | | |
| MCs | −0.863 * | 0.863 * | 1 | | | | | | | | |
| RH | −0.068 | 0.068 | 0.182 | 1 | | | | | | | |
| LCR | 0.059 | −0.059 | 0.095 | −0.273 | 1 | | | | | | |
| DCR | 0.171 | −0.171 | −0.014 | 0.574 * | 0.344 | 1 | | | | | |
| $\sum_{root}$ | 0.124 | −0.124 | −0.208 | 0.315 | −0.198 | 0.047 | 1 | | | | |
| MCr | −0.132 | 0.132 | 0.127 | −0.190 | 0.493 * | −0.033 | −0.071 | 1 | | | |
| $\rho$ | −0.042 | 0.042 | 0.110 | 0.205 | 0.246 | 0.187 | 0.042 | 0.372 | 1 | | |
| V | 0.171 | −0.171 | −0.174 | 0.262 | 0.132 | 0.088 | 0.239 | 0.187 | 0.098 | 1 | |
| $V_{root}$ | 0.323 | −0.323 | −0.228 | −0.152 | 0.238 | 0.099 | −0.505 | −0.005 | −0.137 | 0.453 * | 1 |

Note: *BD*: bulk density; *Po*: soil porosity; *MCs*: soil moisture content; *RH*: soil relative humidity; *LCR*: live crown ratio; *DCR*: mean crown diameter; $\sum_{root}$: number of main roots; *MCr*: root moisture content; $\rho$: woody root biomass density; *V*: velocity; $V_{root}$: velocity of main roots; * Significant at 5% confidence level.

### 3.6. Root Distribution

The root distribution was estimated based on the distribution of the *V* values of the roots in the up-slope and down-slope positions of the soil. The number of $V_{root}$ indicates the number of roots detected in the sample tree. The results show that the average number of roots detected using a root detector was higher than the photogrammetric method (Figures 6–8). The average number of roots detected using a root detector for damar trees (6 ± 2) was higher than for rain trees (5 ± 2), as represented in Table 9. Meanwhile, the photogrammetric method showed the opposite value where the number of damar roots (3 ± 1) was lower than the rain tree (4 ± 1), as shown in Table 9. However, the *t*-test results showed no effect of the tree species on the number of roots in the root detector and photogrammetry methods.

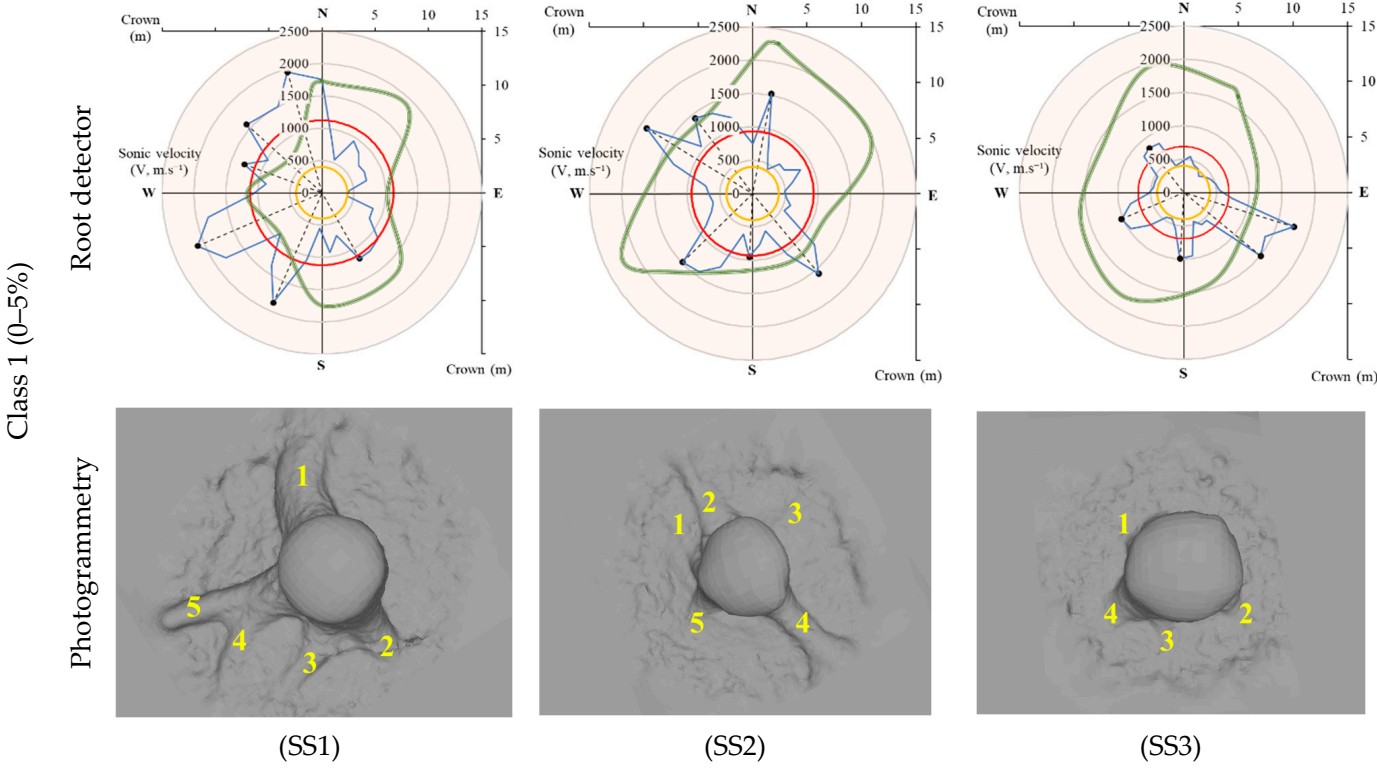

**Figure 6.** *Cont.*

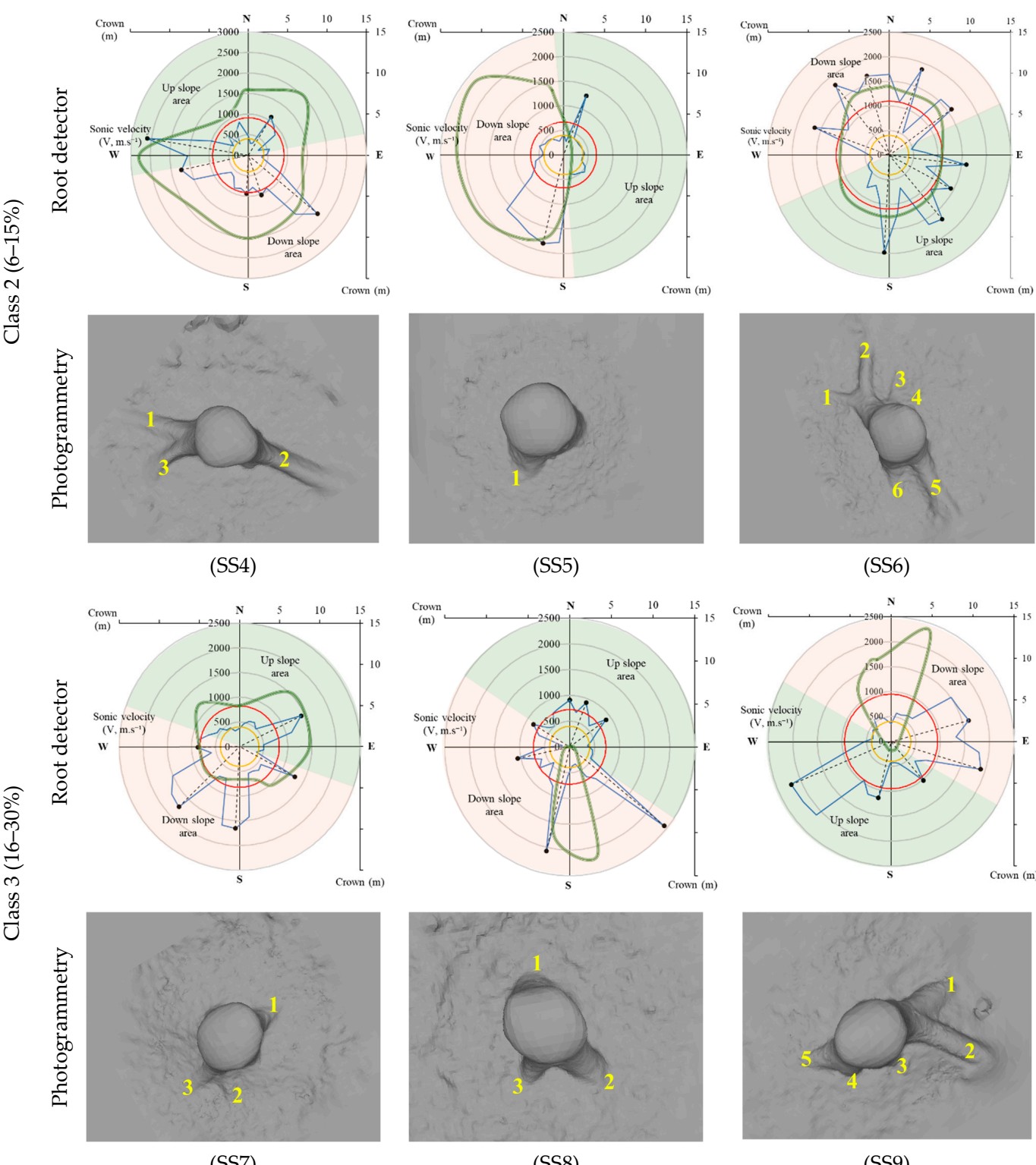

**Figure 6.** *Cont*.

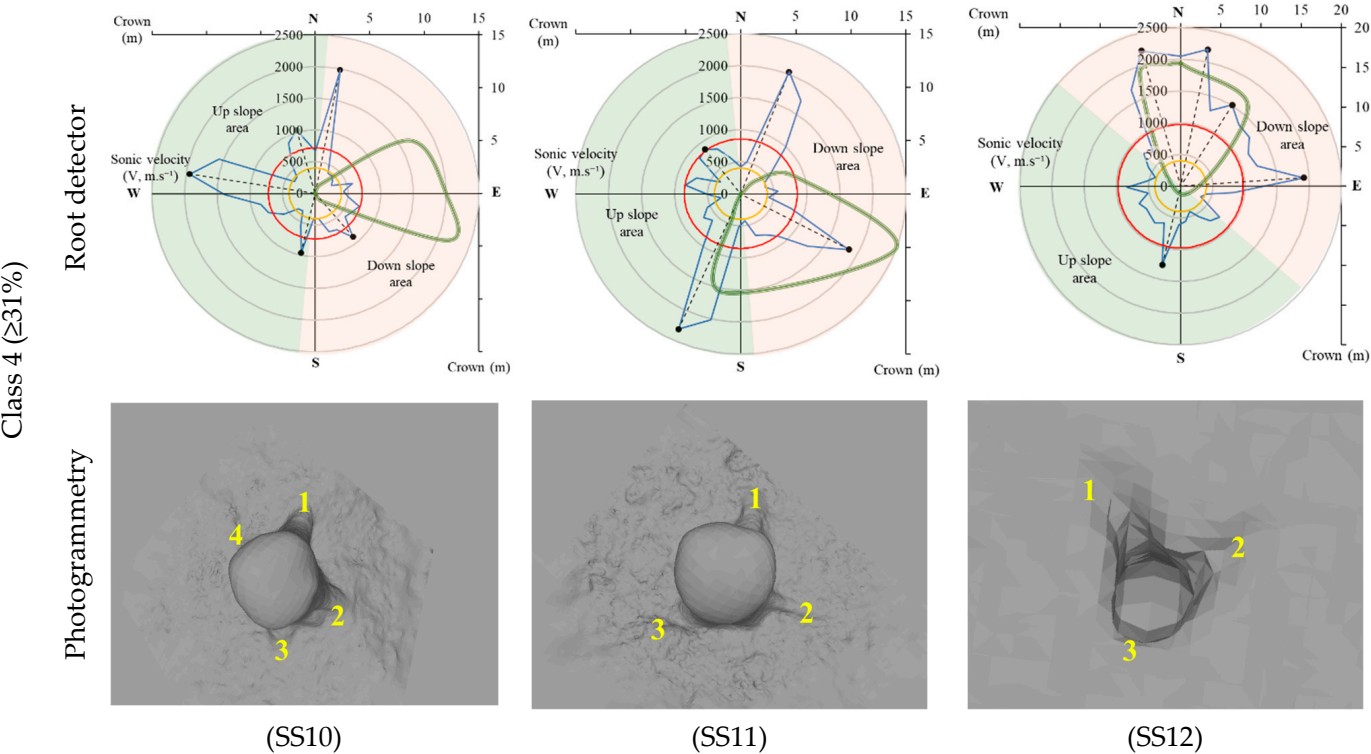

(SS10)    (SS11)    (SS12)

**Figure 6.** Radial root distribution analysis of rain trees (*Samanea saman*) using root detector and visual mapping of the root above ground using the photogrammetry method; classified based on the slope category; class 1 (0–5%): (SS1), (SS2), (SS3); class 2 (6–15%): (SS4), (SS5), (SS6); class 3 (16–30%): (SS7), (SS8), (SS9); and class 4 ($\geq$31%): (SS10), (SS11), (SS12). Sonic velocity in main root noted as "$V_{root}$" and shown as a blue line with the peak in a dark point; sonic velocity generally noted as "$V$" and shown by a blue line; average of $V$ shown in red circle; $V_{root}$ threshold (400 m·s$^{-1}$) in yellow circle.

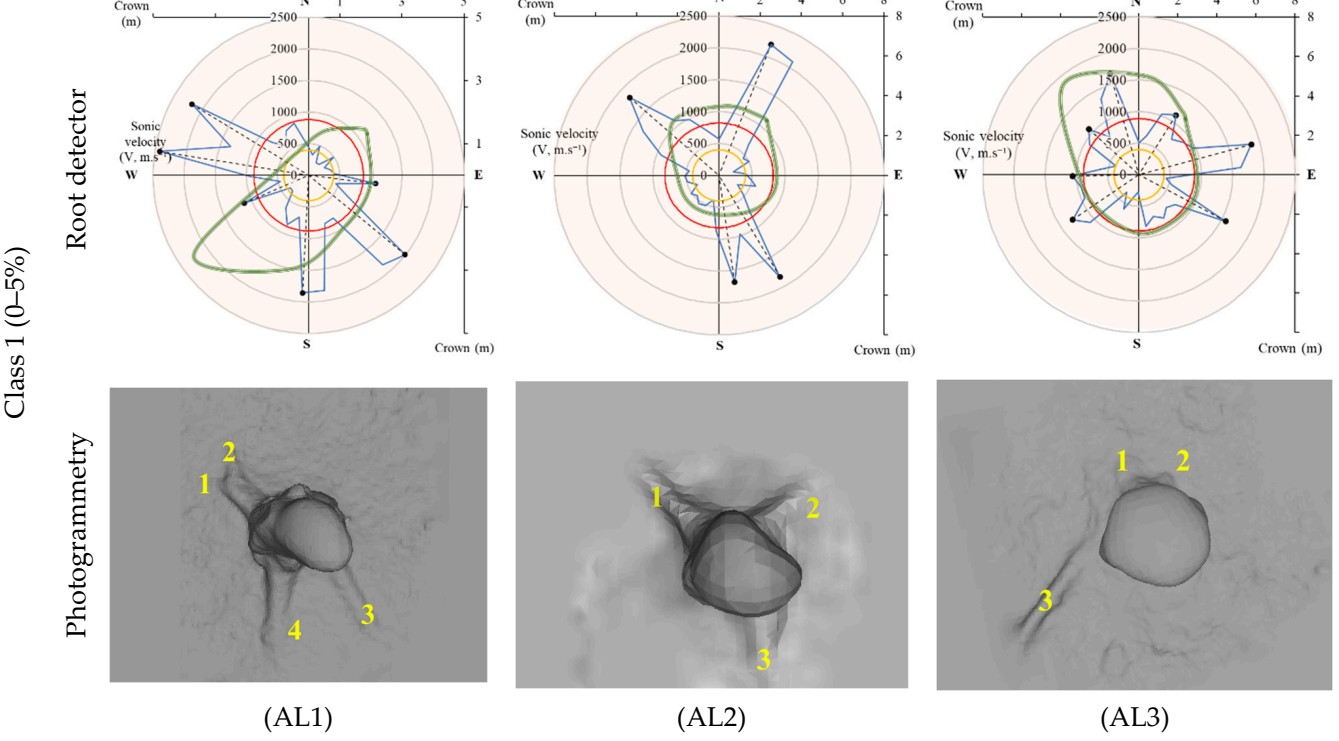

(AL1)    (AL2)    (AL3)

**Figure 7.** *Cont.*

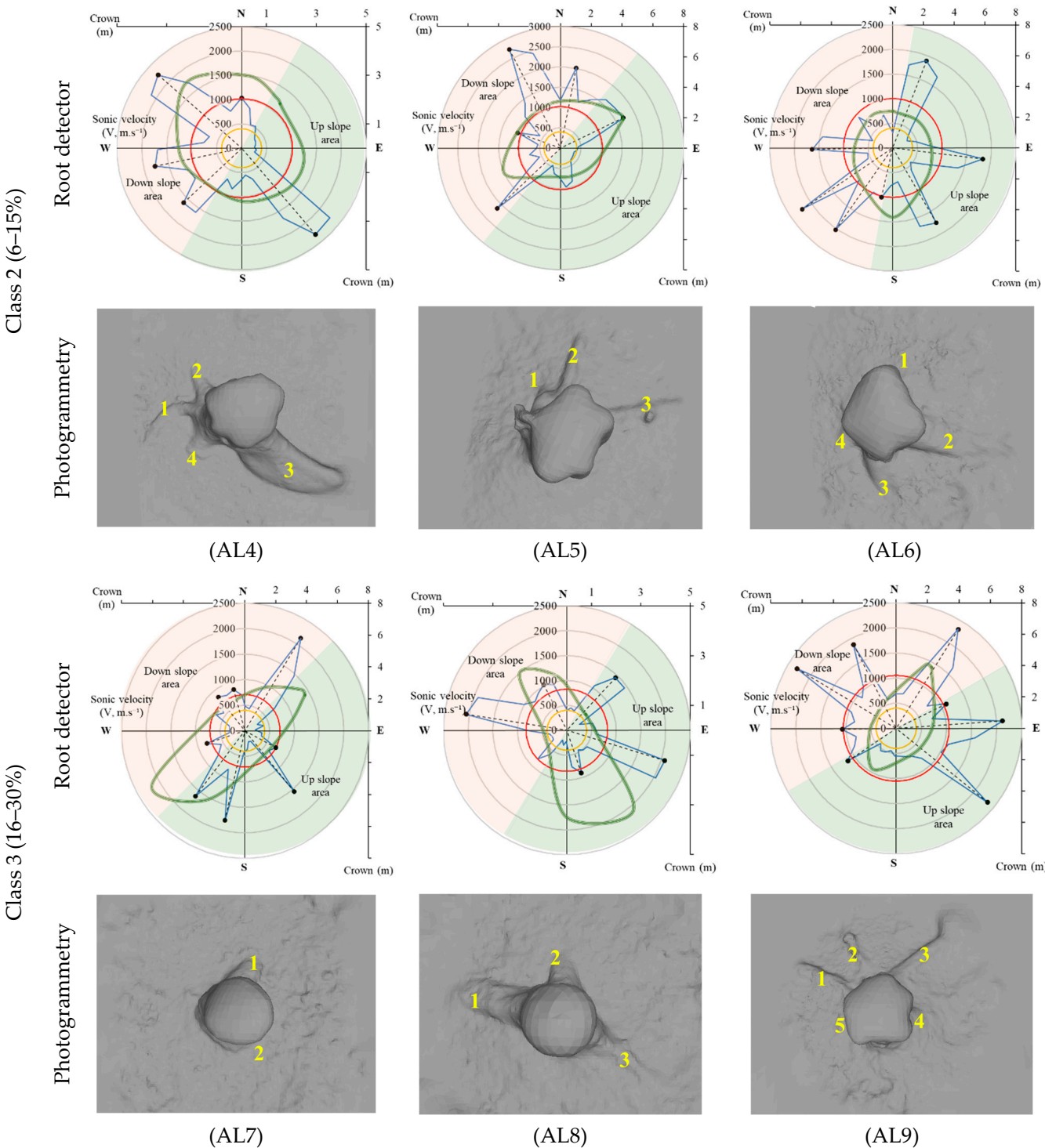

Figure 7. *Cont.*

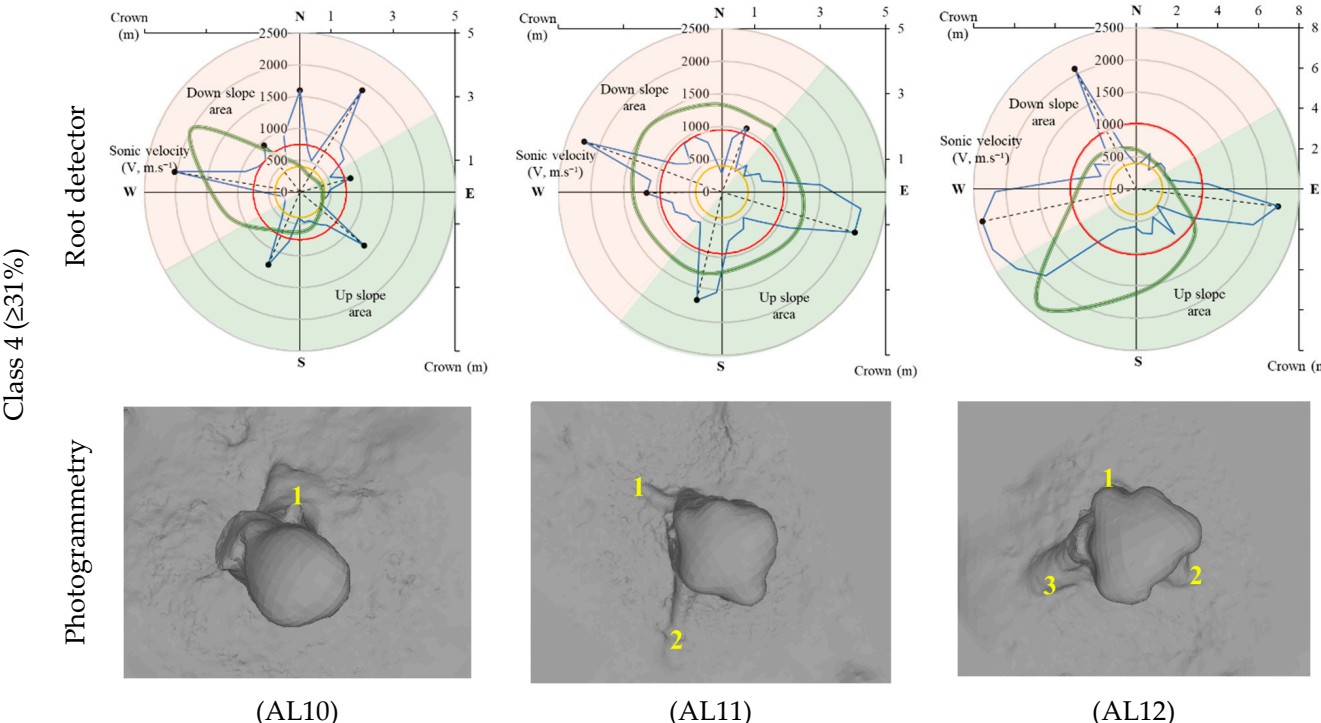

(AL10)  (AL11)  (AL12)

**Figure 7.** Radial root distribution analysis of damar trees (*Agathis loranthifolia*) using the root detector and visual mapping of the root above ground using the photogrammetry method; classified based on the slope category; class 1 (0–5%): (AL1), (AL2), (AL3); class 2 (6–15%): (AL4), (AL5), (AL6); class 3 (16–30%): (AL7), (AL8), (AL9); and class 4 (≥31%): (AL10), (AL11), (AL12). Sonic velocity in main root noted as "$V_{root}$" and shown as a blue line with the peak in a dark point; sonic velocity generally noted as "$V$" and shown by a blue line; average of $V$ shown in red circle; $V_{root}$ threshold (400 m·s$^{-1}$) in yellow circle.

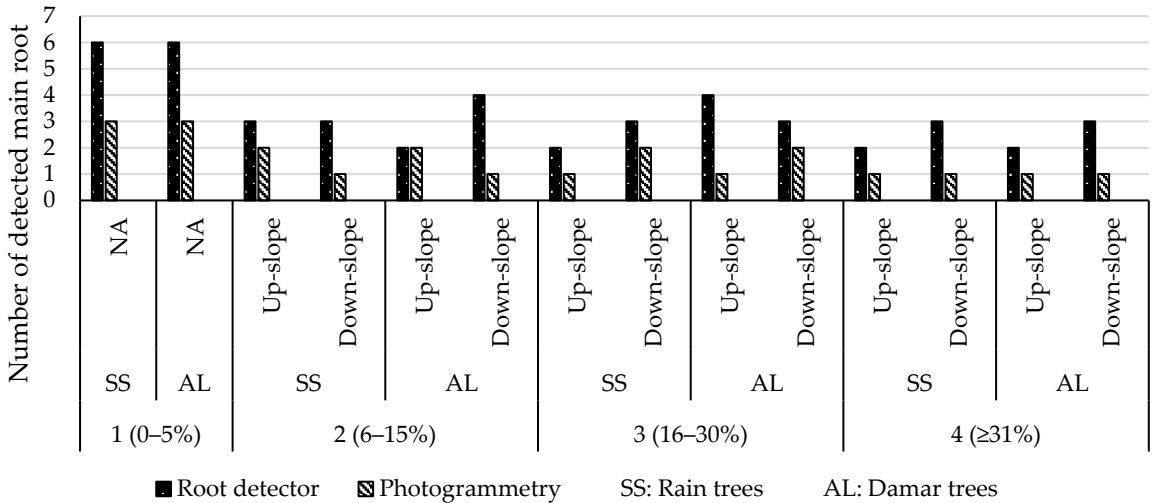

**Figure 8.** The number of main roots ($\Sigma_{root}$) for rain trees and damar trees is based on the class and slope position using the root detector and photogrammetry.

Figures 6 and 7 and Table 12 show the relative direction of the roots toward the tree canopy at various soil slope positions and the number of roots detected using a root detector. Based on the study results, it is known that, in general, the number of roots in the down-slope position is greater than in the up-slope position. Meanwhile, the relationship between the direction of the crown and root distribution shows that it tends to be in the opposite

direction or a distributed condition. Based on the relative orientation of the relationship between the two, in class 1 or flat soil, the concentration of root distribution is opposite to the direction of the crown. Meanwhile, for soil slope classes 2, 3, and 4, most of the root and crown distribution is spread in the down-slope area (Figures 7 and 8).

**Table 12.** The direction of the tree crown and soil slope position in relation to radial root distribution with the number of roots determined by the root detector.

| Soil Slope Class | Tree Code | Soil Slope Position with the Number of Roots ($\sum_{root}$) | | Tree Growth Direction | | Relative Direction Main Roots to Slope/Crown |
|---|---|---|---|---|---|---|
| | | Down Slope | Up Slope | Crown Direction | Root Distribution | |
| 1 (0–5%) | SS 1 | (6) | | NE, SE | NW, SW, SE | Opposite to crown |
| | SS 2 | (5) | | Distributed | Distributed | Opposite to crown |
| | SS 3 | (4) | | Distributed | NW, SE, S, SE | Opposite to crown |
| | AL 1 | (6) | | SW | NW, SW, S, SE | Opposite to crown |
| | AL 2 | (4) | | Distributed | NW, NE, SE | Distributed |
| | AL 3 | (7) | | NW | Distributed | in line to crown |
| 2 (6–15%) | SS 4 | SW, S, E (4) | NW, N, NE (2) | W, S | W, S, SE, NE | Spread in down slope |
| | SS 5 | NW, W, SW (1) | NE, E, SE (1) | W | SW, NE | Opposite to slope |
| | SS 6 | W, NW, N (5) | S, SE, E (4) | Distributed | Distributed | Spread in down slope |
| | AL 4 | W, NW, N (4) | S, SE, E (1) | Distributed | NW, W, SW, SE | Spread in down slope |
| | AL 5 | W, NW, N (4) | S, SE, E (1) | SW, NE | SW, NW, NE | Spread in down slope |
| | AL 6 | SW, W, NW (4) | NE, E, SE (3) | S | Distributed | Spread in down slope |
| 3 (16–30%) | SS 7 | W, SW, S (4) | N, NE, E (1) | NE, E | SW, S, SE, NE | Spread in down slope |
| | SS 8 | W, SW, S (4) | N, NE, E (3) | S | Distributed | Spread in down slope |
| | SS 9 | W, SW, S (2) | N, NE, E (3) | N | NW, SE, E | Spread in up slope |
| | AL 7 | W, NW, N (4) | S, SE, E (4) | SW, NE | Distributed | in line to slope |
| | AL 8 | W, NW, N (2) | S, SE, E (3) | NW, NE, SE | SE, NW | Spread in up slope |
| | AL 9 | W, NW, N (4) | S, SE, E (4) | NE | Distributed | in line to slope |
| 4 (≥31%) | SS 10 | NW, W, SW (2) | NE, E, SE (3) | E | Distributed | Spread in up slope |
| | SS 11 | NW, W, SW (2) | NE, E, SE (2) | S, SE | Distributed | in line to slope |
| | SS 12 | W, SW, S (4) | N, NE, E (1) | N, NE | N, NE, S | Spread in down slope |
| | AL 10 | W, NW, N (4) | S, SE, E (3) | W, NW | Distributed | Spread in down slope |
| | AL 11 | W, NW, N (4) | S, SE, E (2) | Distributed | Distributed | Spread in down slope |
| | AL 12 | W, NW, N (2) | S, SE, E (1) | SW, S | SW, SE, NW | Spread in down slope |

Note: SS: *Samanea saman*; AL: *Agathis loranthifolia*; N: North; NE: Northeast; E: East; SE: Southeast; S: South; SW: Southwest; W: West; NW: Northwest.

### 3.7. Principal Component Analysis

The determination of the relationship grouping among tree morphometric characteristics, physical soil properties, the physical properties of woody root biomass, sound wave velocity, slope class, and slope position on rain and damar trees was carried out using Principal Component Analysis (PCA) with two main factors. This study has a limitation in its sample size. However, the sample size is still large enough to apply PCA in the analysis [37].

The PCA analysis on rain trees (Figure 9) shows that the two main factors of PCA explain 46% of the total diversity, with the percentage of the first factor (F1) at 27% and the second factor (F2) at 19%. Meanwhile, the PCA analysis of damar trees (Figure 10) showed that the two main PCA factors explained 48% of the total diversity, with the percentage of the first factor (F1) at 29% and the second factor (F2) at 19%. The relationship among parameters is also seen based on the direction of the vector. A vector direction <90° indicates that these parameters are positively correlated. In contrast to the angle >90°, it meant a negative correlation. Meanwhile, at an angle of ±90°, it can be confirmed in a weak correlation.

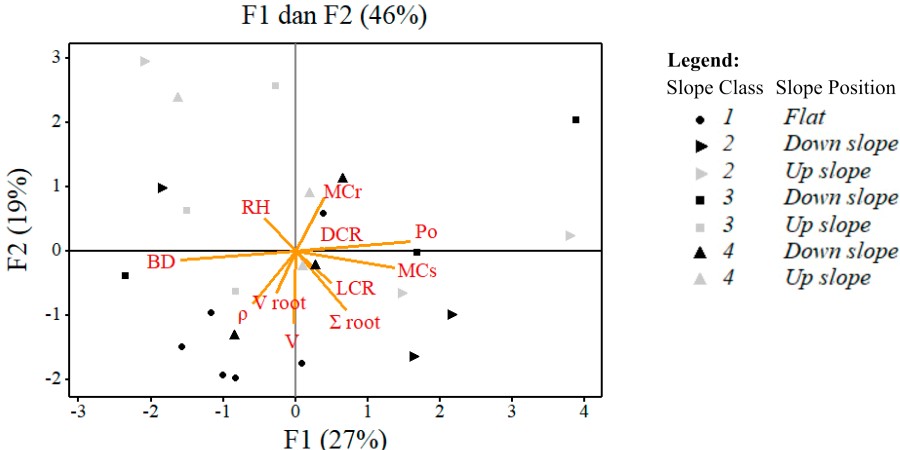

**Figure 9.** Biplot PCA of rain trees (*Samanea saman*).

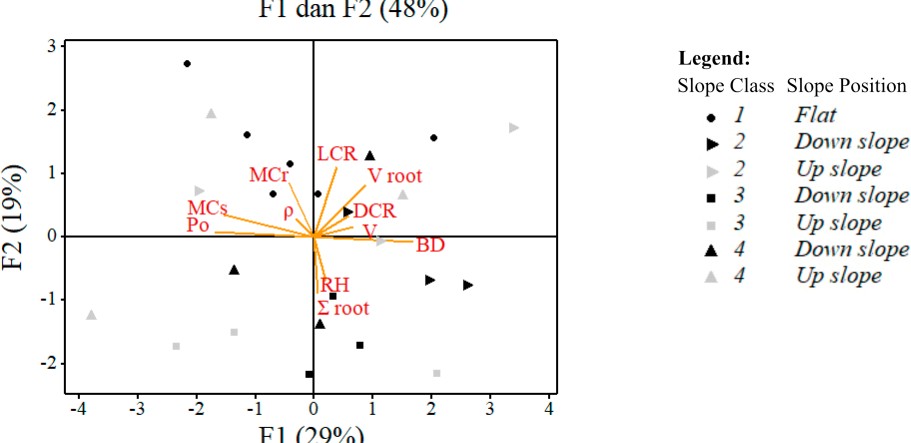

**Figure 10.** Biplot PCA of damar trees (*Agathis loranthifolia*).

The class classification and slope position of the rain trees (Figure 9) show that class 1 rain trees are characterized by the parameters $V$, $V_{root}$, and root density ($\rho$); class 2 is characterized by the parameters of soil moisture content (*MCs*), soil porosity (*Po*), *LCR*, and the number of main roots ($\Sigma_{root}$); class 3 is characterized by bulk density (*BD*), soil relative humidity (*RH*), soil porosity (*Po*), and $V_{root}$; and soil relative humidity (*RH*), root density ($\rho$), *LCR*, number of roots ($\Sigma_{root}$) and root moisture content (*MCr*) characterize class 4. Based on these results, flat soil tends to have higher $V$ and $V$ roots than sloping soil conditions on rain trees. When viewed according to the slope position, the down-slope tends to be characterized by the $V$ parameter compared to the up-slope. It means the down-slope position has a higher $V$ than the up-slope on the rain tree.

The analysis results based on the studied parameters show that parameter $V$ has a negative relationship with the root moisture content (*MCr*) and soil relative humidity (*RH*). Meanwhile, the parameters $V$ to $V_{root}$, root density ($\rho$), the number of roots ($\Sigma_{root}$), and *LCR* have a positive relationship. A weak relationship is shown in the parameters of soil bulk density (*BD*), soil moisture content (*MCs*), soil porosity (*Po*), and *DCR*. This shows that, in the rain trees located on the IPB Campus, the physical properties of the soil, i.e., bulk density (*BD*), soil porosity (*Po*), and soil moisture content (*MCs*), have a weak relationship with the $V$. Only in $V_{root}$, bulk density (*BD*) has a positive relationship. Based on the physical properties of the roots, $V$ has a positive relationship to root density ($\rho$) but has a negative relationship to root moisture content (*MCr*). Meanwhile, the *DCR* morphometric value has a weak relationship, but *LCR* has a positive relationship with $V$.

From the results of the analysis, it was found that class 1 damar trees were characterized by the parameters of the root moisture content (*MCr*), root density ($\rho$), and *LCR*; class 2 is characterized by all parameters; class 3 is characterized by the number of roots ($\Sigma_{root}$) and soil relative humidity (*RH*); all parameters characterize class 4. Based on these results, flat soil tends to have stable *V* and $V_{root}$ compared to sloping soil conditions. When viewed according to the slope position, it is the same as the rain tree. However, in the damar tree, the down-slope position tends to be characterized by the parameter of *V* compared to the up-slope. The down-slope position tends to have a higher *V* than the up-slope.

The analysis results based on the studied parameters show that parameter *V* negatively affects the soil moisture content (*MCs*), root density ($\rho$), and soil porosity (*Po*). Meanwhile, parameter *V* to $V_{root}$, bulk density (*BD*), *DCR*, and *LCR* had a positive relationship. A weak relationship is shown for the parameters of root moisture content (*MCr*), the number of roots ($\Sigma_{root}$), and soil relative humidity (*RH*). This shows that the physical properties of the soil, i.e., bulk density (*BD*), have a positive relationship with *V*, while soil porosity (*Po*) and soil moisture content (*MCs*) have a negative relationship. Based on the physical properties of the roots, *V* has a negative relationship to root density but has a weak relationship to the root moisture content (*MCr*). Meanwhile, *DCR* and *LCR* morphometric values have a positive relationship with *V*.

## 4. Discussion

### 4.1. Tree Morphometric

According to Davis and Jhonson [38], the growth in both diameter and height is profoundly influenced by three environmental factors, encompassing the nutrient content of soil minerals, soil moisture, and sunlight exposure. Additionally, genetic factors play a crucial role, specifically including the genetic balance between the height and diameter growth of a tree. Furthermore, the age of the tree also affects the diameter of the tree, with older trees typically having a larger diameter [39]. Damar trees were higher than rain trees because they have differences in apical dominance. In plants with strong apical dominance, such as damar trees, growth is mostly upright and characterized by a single dominant central axis called the 'excurrent.' On the other hand, rain trees exhibit low apical dominance, resulting in bushy growth, referred to as 'decurrent' [40].

The average crown diameter (*DCR*) of rain trees is higher than damar trees but not significantly different in the live crown ratio (*LCR*). According to Lockhart et al. [41], the shape of a tree crown is influenced by the following two broad factors: the genetic and physical environment. Rain trees have higher a *DCR* than damar trees because damar trees are plants with apical dominance (excurrent), while rain trees develop significant side branches (decurrent). The *LCR* value refers to the part of a tree's crown that has live foliage, so it is considered an indirect measure of a tree's photosynthetic capacity and indicates the competitive status of the trees in a stand. A tree with an *LCR* value of 100% has a maximum leaf surface area and biomass. Meanwhile, an *LCR* value close to 0% indicates that tree growth is hindered due to the limited leaf surface area [42]. Both species have an *LCR* of more than 33%, which indicates that they are suitable for tree growth [43]. The low *LCR* in soil slope class 4 damar trees can occur due to the steep slope position. Additionally, LCR tends to decrease along the greater slope, which is in line with Andrian et al. [44], who stated that sloping land tends to be more easily affected by rainfall, which can cause soil slides so that the fertile topsoil is washed away. The fertile top layer contains soil mineral nutrients that are essential for tree growth [38]. On the steepest slopes, trees grow in the middle slope area, where the terrain is relatively flat. This area has rich soil nutrients and moderate levels of temperature, moisture, and light [45]. These factors might be the cause of why soil class 4 has higher a *DCR* and *LCR* compared to soil slope class 3.

### 4.2. Soil Physic Properties

The research findings indicate that in the up-slope position, the area surrounding rain trees exhibits a higher bulk density (*BD*) value than the down-slope position across

all classes of soil slopes, except for class 2. Conversely, for damar trees, the down-slope position shows a consistently higher *BD* value across all slope classes (refer to Table 2). This variation is attributed to soil texture, influenced by the clay fraction [45]. The disparity in *BD* values between rain trees and damar trees is likely due to the placement of damar trees in more open areas, leading to disturbances and increased soil compaction. Sidewalks and parks in the vicinity further contribute to increased soil compaction. Numerous studies have demonstrated that soil compaction significantly impacts various soil properties, including alterations in the soil structure, an increase in bulk density, heightened penetrometer resistance, diminished soil aeration, reduced water infiltration, and impaired hydraulic conductivity. Additionally, soil compaction poses obstacles to crop growth by impeding root growth mechanically, hampering root architecture, and diminishing the distribution and development of roots [46–52].

Soil porosity (*Po*) is the ratio of nonsolid volume to the total volume of soil [53]. In the tree plantation, soil porosity is important to conduct water, air, and nutrients into the soil [54]. The pore-size distribution provides the ability of soil to store root zone water and air necessary for plant growth [55]. The soil's porosity and pore size distribution directly influence the various soil hydraulic properties, such as hydraulic conductivity, water retention, infiltration, and the available water capacity [56–58].

Soil porosity in damar trees is higher than in rain trees because soil compaction in the IPB Campus area is higher than in GWEF (Table 2). According to Keller [59], soil compaction reduces the volume of a given mass of soil, i.e., a decrease in the void ratio and porosity, which increases the *BD* of soil. The soil moisture content (*MC*) has a dominant influence on root growth through the direct effects of water availability on root growth, the effects of water on photosynthesis and, therefore, carbohydrate availability, the effects of water on oxygen availability in wet soils, and the effects of soil impedance on root growth because dry soils tend to be hard [60].

The slope influences the soil moisture content because of the aspect associated with the soil's relative humidity (*RH*), which is the water stored between the soil's pores. Dynamic soil humidity is caused by evaporation through the soil surface, transpiration, and percolation [61]. Low soil humidity in soil class 1 damar trees is thought to be due to a low crown density and lack of litter on the soil surface. Reduced crown density can cause the sun's heat to be more easily exposed directly to the ground, increasing the soil temperature. An increase in soil temperature can increase the evaporation process so that the water content in the soil decreases [62]. There was no significant difference between rain trees and damar trees regarding the slope position. This result is in line with Solgi & Nafaji [63], who stated that, in the sloping class bulk density, porosity might not be significant except if it is disturbed. Therefore, the location and slope could also affect these properties [64].

### 4.3. Physical Properties of Woody Root Biomass

According to Guo et al. [65], the moisture content in trees can be influenced by many factors, including the place where its roots age, soils, and seasons, and it impacts its strength [66]. Damar trees grow in the forest environment and have a higher soil moisture content, so the roots have a greater water supply than rain trees, which grow in urban areas. Another study said that a low availability of water leaves trees with inadequate access to water. It can reduce the transpiration rate in the long term [67].

This study found no significant difference in the fresh wood root biomass density to the soil slope class and position in both tree species. Wood density is influenced by the following several factors: the species of the tree [68], ages [69], the growth conditions of the soil [70], topography [71,72], and inter-tree competition [73]. According to Mahajan [74], the density of wood depends on the weight of water in a given volume of wood, the weight of the wood substance in a given volume of wood, and the volume of wood at a specified moisture content. However, root density did not directly affect the permeability, which affects the root's moisture content. Rain trees have more complicated anatomical

features and greater structural variation than damar trees, resulting in a greater range of permeability and capillary behavior [75,76].

### 4.4. Root Detection and Distribution

The roots' presence is evaluated by detecting roots through the pattern of root distribution in the soil using a root detector. The parameter used is sonic wave velocity through wave propagation. Oliveira et al. [77] and Bucur [35] mentioned that the high moisture content in wood tends to slow down the speed of wave propagation. In this study, sound waves do not only propagate through the wood (root mass) but also in the soil, so the ground train also affects the value of the sonic velocity ($V$). Apart from the water content of the propagation medium, other factors, such as the anatomical structure. can also influence the speed of propagation of sound waves in wood [35].

Variations in the average $V_{root}$ value can be influenced by various factors such as the woody biomass moisture content, the direction of fiber, fiber length, cell wall, crystalline region composition, and growth circle structure [35,77]. The difference between species is presumably because the anatomical structure of the damar tree (softwood) is more ho-mogeneous than that of the rain tree (hardwood). According to research conducted by Karlinasari et al. [78], the speed of sound waves propagating in pine (softwood) is higher than in rosewood (hardwood). This is because pine (softwood) has a homogeneous cell structure, long fibers, low porosity, permeability of the cell wall, small microfibrils (more parallel fiber directions), and larger crystalline regions. This study is also a reason for the positive correlation between the moisture contents ($MCr$ and $MCs$). The positive correlation also can be caused by the distance between the root and the tools, as described by Proto et al. [1].

Comparing $\sum_{root}$ based on two methods, the number of roots based on photogrammetry is always below the root detector. According to Rahman et al. [21], the photogrammetric method cannot fully describe the distribution of roots because underground roots cannot be visually detected. However, this method is useful for providing an overview of the distribution of radial roots when excavation methods are not possible. Environmental soil conditions also affect the resulting image. In addition, the photogrammetry method only captures the shallow roots that are visible on the surface, while root detectors can detect both above and underground roots. The results of the photogrammetry and root detector tests showed that the rain tree roots were shallower [79], as they could be visually detected, while the damar tree roots were deeper.

In addition, Rahman et al. [28] stated that large root sizes can affect the distribution of roots detected by the root detector. In one large root, 2–3 points can be detected due to the high $V$ value of the roots, causing the number of roots detected based on the $V_{root}$ (root detector) to be higher than those that appear on the surface (photogrammetry). These results indicate that photogrammetry is only used to validate additional data on root direction from the root detector. The number of roots resulting from photogrammetry cannot be used to describe the root distribution, even for the number of roots. The number of roots is important for tree stability, especially the main root. Primarily, in the case of wind-thrown beech trees, the root with a diameter >10 cm ranges between 3 and 22% of the total root number in root systems but is the most important for tree stability [6].

The root distribution or number of roots is greater on the down-slope than on the up-slope. This can be caused by differences in the soil's physico-chemical properties [80]. In some systems, more roots are oriented down-slope than up-slope due to gravitropism [81]. Roots oriented down-slope convey water more efficiently [81]. However, there are three trees that have more roots up-slope; it might be that some systems have more roots growing up-slope from the stem, which suggests that some roots may undergo hydrotropism rather than gravitropism if more water is located up-slope [82].

The results show that the distribution of roots tends to be in the same direction as the crown (Figures 6 and 7). The size and shape of the tree canopy can change due to variations in age, where it grows with environmental conditions, competition, and plant

spacing [83,84]. According to Stokes et al. [85] and Chiatante et al. [61], environmental conditions, especially wind direction and soil slope, can affect root distribution. Trees with a symmetrical shape tend to have a distribution of roots in all directions. Meanwhile, an asymmetrical root distribution can be caused by environmental and mechanical stress as a form of tree response in increasing its stability through root distribution. Based on their spread direction (Table 12), root distribution showed that the down-slope has a more aligned direction with radial root distribution than the up-slope. The crown direction tends to be opposite to the radial root distribution. According to Rahman et al. [28], although roots may not be detected in the direction of the tree crown load, this does not mean that there is no root, especially if there is a vertical root (sinker root).

## 5. Conclusions

Based on this study, the sound wave propagation detected as root biomass can be used to determine the number of main roots, which was validated using the photogrammetry method as a visual image of the main roots visible on the soil surface. Through the angular data analysis of the sound wave propagation of the root ($V_{root}$), it was found that the soil slope class can influence the $V_{root}$ value. Furthermore, this research revealed that the down-slope position of the tree tended to have more roots than the up-slope position. This can be explained by the fact that trees can respond to sloping soil conditions, where increased root distribution enhances tree stability.

An examination of the relationship between crown projection, root distribution, soil slope class, and slope position showed that the down-slope position of the tree exhibited a more aligned direction with the radial root distribution in terms of the crown growth direction and compared to the up-slope. The results of the Principal Component Analysis (PCA) indicated that the clustering of data was more closely associated with the slope position than with the slope class. Further studies, mathematical analysis, and simulations are very interesting and helpful in answering the variations in environmental influences on the presence of roots.

**Author Contributions:** Conceptualization, L.K. and M.M.R.; methodology, U.D.S., M.M.R. and L.K.; validation, U.D.S., L.K. and I.Z.S.; data curation, U.D.S.; writing—original draft preparation, M.T.; writing—review and editing, L.K. and M.M.R.; visualization, M.T. All authors have read and agreed to the published version of the manuscript.

**Funding:** This research was funded by Directorate General Higher Education, Riset, and Technology, Ministry of Education, Culture, Research, and Technology, grant number 001/E5/PG.02.00.PL/2023 and 15856/IT3.D10/PT.01.02/P/T/2023 for Forest Biomechanics research in the scheme of Basic Research of Higher Education.

**Data Availability Statement:** The data presented in this study are available on request from the corresponding author.

**Acknowledgments:** We thank Gunung Walat University Forest Management for permission and support on field research.

**Conflicts of Interest:** The authors declare no conflict of interest.

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
