# Peer review of "Clarifying the Main Root Distribution of Trees in Varied Slope Environments Using Non-Destructive Root Detection"

_forests, doi:10.3390/f14122434_

Round 1

Reviewer 1 Report

Comments and Suggestions for Authors

Comments:

The manuscript titled “Assessing Radial Root Distribution: Insights from Excurrent and Decurrent Trees in Varied Slope Environments Using Non-Destructive Root Detection” by Taufiqurrachman et al. is interesting. It elaborates on the usefulness of audiometric method to detect radial roots. However, there are some points that needs to be addressed. Authors are requested to go through the comments listed below.

Major comments:

1.     The authors took just two tree species Agathis loranthifolia and Samanea saman as the respective representatives of the excurrent and decurrent tree types. Even though the study was mainly for the validation/confirmation of the usefulness of the instrument, the study would have benefitted more had the authors used at least three species for each tree types. Authors are kindly suggested to do so, if feasible.

2.     The audiometric method appears to give relative higher number/accuracy of radial roots on/near surface as compared to the photometric method. Would it be possible to retrieve information on additional radial roots if the receiver node of the instrument were to place at certain depth rather than placing it in the surface? Such case may have provided the data that would otherwise have been impossible to retrieve via the photometric method.

Minor comments:

1.     Please include the sample numbers as well as the symbol/acrony definitions at tables/figures wherever relevant.

Author Response

No

Comment

Response

1

The authors took just two tree species Agathis loranthifolia and Samanea saman as the respective representatives of the excurrent and decurrent tree types. Even though the study was mainly for the validation/confirmation of the usefulness of the instrument, the study would have benefitted more had the authors used at least three species for each tree types. Authors are kindly suggested to do so, if feasible

This preliminary study is limited by species and resources, so further research will be conducted in a different study.

2

The audiometric method appears to give relative higher number/accuracy of radial roots on/near surface as compared to the photometric method. Would it be possible to retrieve information on additional radial roots if the receiver node of the instrument were to place at certain depth rather than placing it in the surface? Such case may have provided the data that would otherwise have been impossible to retrieve via the photometric method.

The photometric method is a visual technique that only represents the surface roots. On the other hand, the root detector can detect the main root that lies beneath the soil. Although it can only detect up to a depth of 1 meter, it is sufficient as the main roots do not go very deep.

3

Please include the sample numbers as well as the symbol/acrony definitions at tables/figures wherever relevant.

We already put the number of samples (n) and put symbol definition for each table or graph

Reviewer 2 Report

Comments and Suggestions for Authors

Dear colleagues!

The article “Assessing Radial Root Distribution: Insights from Excurrent and Decurrent Trees in Varied Slope Environments Using Non-Destructive Root Detection” is a methodical study, but this research has a small sample of data.

The list of comments:

1. The section “Introduction”. Line 43: “…and drainage).”; Line 47: “…stand density).”; Line 65: “...crown characteristics.”. References to literature cited are required.

2. The section “Introduction”. Line 48-50: paraphrase a sentence from the definition, wording to the description of the identified functions.

3. It is necessary to strengthen the relevance of the research in the section in the section “Introduction".

4. The section “Materials and methods”. In this study that is methodical research, two methods (“Root Detector” and “Photogrammetry”) are compared, but the sample is insufficient, especially for the correct using the multidimensional analysis methods, such as the principal component analyses, PCA (Line 215). Number of plant species: 2. Number of plants: 3 for each class of slope. Total number of plants analyzed: 24. Number of variables in PCA: 11 (see Figures 8-9). PCA (Lines 477-536) can only be used if the sample of the data is increased by an order of magnitude.

5. The section “Materials and methods”. Line 84: “GWEF is located at an altitude of 460-715 meters above 84 sea level with an area of 359 ha.” It is unclear how the representativeness of the sample was achieved.

6. The section “Materials and methods”. Line 103. Correct: “...class 316-30%),”.

7. The section “Materials and methods”. It is unclear why the authors investigated only main roots, which are few. An increase in the sample would be possible by analyzing the roots of the following orders, if it is possible.

8. The section “Results and Discussion”. The text is hard to read, because the results and the discussion are combined in one section.

9. The section “Results and Discussion”. Line 230 and further: “(50.41±28.38)%”, it is necessary to remove the brackets and standard errors, since it is already presented in Table 2.

10. The section “Results and Discussion”. Figure 5 (Line 252): add statistics (paired t-test), remove signatures – digits of parameter values above the columns.

11. The section “Results and Discussion”. Lines 276-277: “[25], [26], [27], [28], [29], [30], [31], [32], and [33]” replace with “[25-33]". It is necessary to reduce the number of references to the predictable consequence of soil compactions.

12. The section “Results and Discussion”. Table 2, Line 416: “0.00*”. Increase the dimension.

13. The section “Results and Discussion”. Lines 485-486, 489: “This means that there is still 54% of the variance that biplot graphs cannot explain.” and “This means there is still 52% of the variation that biplot graphs cannot explain.”. Paraphrase: most of the variance cannot be explained by the two principal components.

14. The section “Conclusion”. Line 552: it is necessary to indicate the practical significance of the results and conclusions obtained. Judging by what the article states in the section “Materials and methods” (Line 188-191), the proposed method of root analysis is difficult to scale.

15. The section “Conclusion”. Line 539-541: “There is no significant difference in the distribution of the roots of rain trees and damar trees concerning the slope class of the soil.” Paraphrase: this conclusion is true only for main roots.

16. Check the meaning of phrases and English translation: “sloping soil” (Line 61); Lines 66-68; “Physics" (Line 137); “coarse" (Line 170); “tree. furthermore,” (Line 227); “Descriptively,” (Line 226, 287); “(iii)” (Line 276); “The slope aspect” and “the aspect” (Lines 303-304).

Reviewer's conclusion: reconsider after major revision.

Comments on the Quality of English Language

16. Check the meaning of phrases and English translation: “sloping soil” (Line 61); Lines 66-68; “Physics" (Line 137); “coarse" (Line 170); “tree. furthermore,” (Line 227); “Descriptively,” (Line 226, 287); “(iii)” (Line 276); “The slope aspect” and “the aspect” (Lines 303-304).

Author Response

No

Comment

Response

1. The section “Introduction”. Line 43: “…and drainage).”; Line 47: “…stand density).”; Line 65: “...crown characteristics.”. References to literature cited are required.

We have already put the citation in those sentences.

2. The section “Introduction”. Line 48-50: paraphrase a sentence from the definition, wording to the description of the identified functions.

We already paraphrase the sentence.

3. It is necessary to strengthen the relevance of the research in the section in the section “Introduction".

The introduction has been updated

4. The section “Materials and methods”. In this study that is methodical research, two methods (“Root Detector” and “Photogrammetry”) are compared, but the sample is insufficient, especially for the correct using the multidimensional analysis methods, such as the principal component analyses, PCA (Line 215). Number of plant species: 2. Number of plants: 3 for each class of slope. Total number of plants analyzed: 24. Number of variables in PCA: 11 (see Figures 8-9). PCA (Lines 477-536) can only be used if the sample of the data is increased by an order of magnitude.

PCA is used to evaluate main variables due to same proxy. It can reduce variables in future studies.

5. The section “Materials and methods”. Line 84: “GWEF is located at an altitude of 460-715 meters above 84 sea level with an area of 359 ha.” It is unclear how the representativeness of the sample was achieved.

The altitude is just an indicator of the site area, and does not represent the sampling. We have already removed this information.

6. The section “Materials and methods”. Line 103. Correct: “...class 316-30%),”.

We have made a correction to the data. The correct value should be 3 and the percentage range is 16-30%.

7. The section “Materials and methods”. It is unclear why the authors investigated only main roots, which are few. An increase in the sample would be possible by analyzing the roots of the following orders, if it is possible.

The tool only capable to detect the large root

8. The section “Results and Discussion”. The text is hard to read, because the results and the discussion are combined in one section.

We have divide result and discussion become different section.

9. The section “Results and Discussion”. Line 230 and further: “(50.41±28.38)%”, it is necessary to remove the brackets and standard errors, since it is already presented in Table 2.

We already deleted the standard deviation for each data in the text

10. The section “Results and Discussion”. Figure 5 (Line 252): add statistics (paired t-test), remove signatures – digits of parameter values above the columns.

Data label has been deleted, the t-test between group has been conducted

The section “Results and Discussion”. Lines 276-277: “[25], [26], [27], [28], [29], [30], [31], [32], and [33]” replace with “[25-33]". It is necessary to reduce the number of references to the predictable consequence of soil compactions.

We already corrected the reference and citation style

The section “Results and Discussion”. Table 2, Line 416: “0.00*”. Increase the dimension.

Thank you, we have revised for the digit.

The section “Results and Discussion”. Lines 485-486, 489: “This means that there is still 54% of the variance that biplot graphs cannot explain.” and “This means there is still 52% of the variation that biplot graphs cannot explain.”. Paraphrase: most of the variance cannot be explained by the two principal components.

Thank you, we already re-phrase with your suggested sentence

14. The section “Conclusion”. Line 552: it is necessary to indicate the practical significance of the results and conclusions obtained. Judging by what the article states in the section “Materials and methods” (Line 188-191), the proposed method of root analysis is difficult to scale.

The conclusion has been revised and updated.

15. The section “Conclusion”. Line 539-541: “There is no significant difference in the distribution of the roots of rain trees and damar trees concerning the slope class of the soil.” Paraphrase: this conclusion is true only for main roots.

"Thank you for your suggestion. We have already rephrased the text using your suggested sentence."

16. Check the meaning of phrases and English translation: “sloping soil” (Line 61); Lines 66-68; “Physics" (Line 137); “coarse" (Line 170); “tree. furthermore,” (Line 227); “Descriptively,” (Line 226, 287); “(iii)” (Line 276); “The slope aspect” and “the aspect” (Lines 303-304).

The phrases have already been changed.

Reviewer 3 Report

Comments and Suggestions for Authors

Generic

The test of the 'sonic root detector' under other circumstances than for which it has been developed is welcome. Questions of root orientation on slopes are relevant -- although there are alternative hypotheses on functional aspects (tree mechanical stability, water & nutrient uptake opportunities) that are only hinted at here, rather than analyzed.

The excurrent-decurrent contrast that's in the title would require a substantially larger data set (more tree species) for testing rather than 1 species in each class. Please leave it out of title and abstract; other differences, such as wood density, offer more direct explanations for your results.

The method descriptions that you refer to suggest the receptor sensors to be placed at 6 x DBH, rather than a standardized 80 cm; you'll need to justify this deviation.

Information on DBH is missing -- yet, it has been used as a scaling parameter for many tree properties, including proximal root data -- as in a study on Indonesian trees on sloping land:

Hairiah, K., Widianto, W., Suprayogo, D. and Van Noordwijk, M., 2020. Tree roots anchoring and binding soil: Reducing landslide risk in Indonesian agroforestry. Land, 9(8), p.256.

Specific

Title could be: Coarse root distribution of trees on slopes clarified by a sonic root detector

Introduction: The first two sentences can be omitted, as they hardly prepare for what follows.

Introduction might position the 'sonic root detector' in a tradition of studying electrical conductance (Chloupek, O., 1977. Evaluation of the size of a plant's root system using its electrical capacitance. Plant and Soil, 48(2), pp.525-532. 

Dalton, F.N., 1995. In-situ root extent measurements by electrical capacitance methods. Plant and soil, 173, pp.157-165.  

Dietrich, R.C., Bengough, A.G., Jones, H.G. and White, P.J., 2013. Can root electrical capacitance be used to predict root mass in soil?. Annals of Botany, 112(2), pp.457-464.)

Ellis, T.W., Murray, W., Paul, K., Kavalieris, L., Brophy, J., Williams, C. and Maass, M., 2013. Electrical capacitance as a rapid and non-invasive indicator of root length. Tree physiology, 33(1), pp.3-17.)

and ground-penetrating radar (Hruska, J., Čermák, J. and Šustek, S., 1999. Mapping tree root systems with ground-penetrating radar. Tree physiology, 19(2), pp.125-130.

Guo, L., Chen, J., Cui, X., Fan, B. and Lin, H., 2013. Application of ground penetrating radar for coarse root detection and quantification: a review. Plant and soil, 362, pp.1-23.

Butnor, J.R., Doolittle, J.A., Johnsen, K.H., Samuelson, L., Stokes, T. and Kress, L., 2003. Utility of ground‐penetrating radar as a root biomass survey tool in forest systems. Soil Science Society of America Journal, 67(5), pp.1607-1615.) summarizing strengths and weaknesses of both methods. The sonic root detector appears to be easier and lower-cost, with less sensitivity to soil water content (and no sensitivity to electrolyte contents that hamper interpretation of the electical conductance   data)

Line 89 'choppy'? please check classification system used; more relevant is the actual slope angle on which the trees grew.

Line 140 How were the destructive root samples collected?

Section 2.5: more detail is needed on how the method was used and how data are interpreted (even if the details may be part of proprietary software): please outline the principles used 

Figure 3C: the dotted lines connecting tree center to local maxima in sonic velocity are probably not the most likely directions of roots... Have alternative methods of interpretation been tested?

Line 202: what does this mean? please reformulate

Section 2.7 Beyond the software used, the reader will want to know what level of 'independence' and 'replication' was tested, as the confounding of species x location could not be unpacked. How did you construct 'random' null hypotheses to be rejected (or not)? The angular distribution has some specific challenges. ANOVA application needs more detail.

3. Results and Discussion: please separate into two sections, and first give the results, before discussing your questions

Line 218: this is not a result, but a direct part of your sample selection. You need to clarify the criteria you used for inclusion in the sample.

Line 219: given this tee diameter, your deviation from the 6xDBH advise is very considerable...

Line 221 "The t-test results showed that tree species significantly affected the average value of tree height" NO! the t-test shows that it is unlikely that the two tree species belong to a homogeneous distribution, no causality implied 

Fig 5 This belongs in a 'sample characterization' part of the methods (clarifying how sample selection rules worked out)

For all 'upslope' versus 'downslope' comparisons as need a much stronger statistical analysis of angular distributions

-- I stopped reviewing here, as you'll need more convincing statistical treatment of your data

Comments on the Quality of English Language

Generally understandable

Author Response

No

Comment

Response

The test of the 'sonic root detector' under other circumstances than for which it has been developed is welcome. Questions of root orientation on slopes are relevant -- although there are alternative hypotheses on functional aspects (tree mechanical stability, water & nutrient uptake opportunities) that are only hinted at here, rather than analyzed.

The excurrent-decurrent contrast that's in the title would require a substantially larger data set (more tree species) for testing rather than 1 species in each class. Please leave it out of title and abstract; other differences, such as wood density, offer more direct explanations for your results.

Tree mechanical properties were represented by LCR and DCR, while environment parameters were captured by RH, Rho, dan MC. All those parameters were analyzed based data that we have.

Thank you for your suggestion. We have taken it into consideration and updated our title accordingly.

The method descriptions that you refer to suggest the receptor sensors to be placed at 6 x DBH, rather than a standardized 80 cm; you'll need to justify this deviation

Based on our previous study on mature trees and the reference study conducted by Proto et al, it has been determined that when using small plants, the maximum distance for consistent velocity is 200cm. However, at 120cm, the velocity is not consistent. After analyzing both our study and the reference, it has been concluded that the optimum distance for this method is 80cm.

Information on DBH is missing -- yet, it has been used as a scaling parameter for many tree properties, including proximal root data -- as in a study on Indonesian trees on sloping land:

Hairiah, K., Widianto, W., Suprayogo, D. and Van Noordwijk, M., 2020. Tree roots anchoring and binding soil: Reducing landslide risk in Indonesian agroforestry. Land, 9(8), p.256.

Tree stability on DCR has been established by using DBH as a proxy. Additionally, the reference has been added for further clarification.

Title could be: Coarse root distribution of trees on slopes clarified by a sonic root detector

Based on your suggestion, we have changed the title.

Introduction: The first two sentences can be omitted, as they hardly prepare for what follows.

Introduction might position the 'sonic root detector' in a tradition of studying electrical conductance (Chloupek, O., 1977. Evaluation of the size of a plant's root system using its electrical capacitance. Plant and Soil, 48(2), pp.525-532. 

Dalton, F.N., 1995. In-situ root extent measurements by electrical capacitance methods. Plant and soil, 173, pp.157-165.  

Dietrich, R.C., Bengough, A.G., Jones, H.G. and White, P.J., 2013. Can root electrical capacitance be used to predict root mass in soil?. Annals of Botany, 112(2), pp.457-464.)

Ellis, T.W., Murray, W., Paul, K., Kavalieris, L., Brophy, J., Williams, C. and Maass, M., 2013. Electrical capacitance as a rapid and non-invasive indicator of root length. Tree physiology, 33(1), pp.3-17.)

and ground-penetrating radar (Hruska, J., Čermák, J. and Šustek, S., 1999. Mapping tree root systems with ground-penetrating radar. Tree physiology, 19(2), pp.125-130.

Guo, L., Chen, J., Cui, X., Fan, B. and Lin, H., 2013. Application of ground penetrating radar for coarse root detection and quantification: a review. Plant and soil, 362, pp.1-23.

Butnor, J.R., Doolittle, J.A., Johnsen, K.H., Samuelson, L., Stokes, T. and Kress, L., 2003. Utility of ground‐penetrating radar as a root biomass survey tool in forest systems. Soil Science Society of America Journal, 67(5), pp.1607-1615.) summarizing strengths and weaknesses of both methods. The sonic root detector appears to be easier and lower-cost, with less sensitivity to soil water content (and no sensitivity to electrolyte contents that hamper interpretation of the electical conductance   data)

Thank you for the suggested references. We have considered and cited them. Additionally, we have revised the introduction.

Line 89 'choppy'? please check classification system used; more relevant is the actual slope angle on which the trees grew

In the reference, there is no detail about the classification. We rephrased the sentence to describe that the location has varying slopes and elevations.

Line 140 How were the destructive root samples collected?

As per the methodology, we obtained small root samples, each measuring 2x2x1 cm, from two primary roots - one from the up-slope and the other from the downslope. In order to determine the root density, two samples were collected from each tree, and the Archimedes method was used.

Section 2.5: more detail is needed on how the method was used and how data are interpreted (even if the details may be part of proprietary software): please outline the principles used 

Section 2.5 now provides a more comprehensive explanation of the tool's working principle.

Figure 3C: the dotted lines connecting tree center to local maxima in sonic velocity are probably not the most likely directions of roots... Have alternative methods of interpretation been tested?

To uncover the coarse roots, we rely on visual methods, specifically photogrammetry, as illustrated in Figure 3b. Proto's 2020 study tested root detection through excavation, but it was found that the direction of detected roots could provide proximity to the actual root direction, rather than being exact.

Line 202: what does this mean? please reformulate

The sentence regarding the range of slopes within the same classification has been revised to be more clear.

Section 2.7 Beyond the software used, the reader will want to know what level of 'independence' and 'replication' was tested, as the confounding of species x location could not be unpacked. How did you construct 'random' null hypotheses to be rejected (or not)? The angular distribution has some specific challenges. ANOVA application needs more detail.

We utilized angular distribution in our statistical analysis and repeated the ANOVA.

3. Results and Discussion: please separate into two sections, and first give the results, before discussing your questions

We separate it into two sections

Line 218: this is not a result, but a direct part of your sample selection. You need to clarify the criteria you used for inclusion in the sample.

We moved that part in the method

Line 219: given this tee diameter, your deviation from the 6xDBH advise is very considerable...

We agree that the best distance to identify root biomass is at 6xDBH. However, the root detector cannot detect roots beyond 200 cm from the trunk center. Proto et al established that the 6xdbh threshold applies only to trees up to 120, rendering it inapplicable to larger trees.

Line 221 "The t-test results showed that tree species significantly affected the average value of tree height" NO! the t-test shows that it is unlikely that the two tree species belong to a homogeneous distribution, no causality implied 

We have made changes to the sentences and conducted a t-test to confirm differences in tree species characteristics.

Fig 5 This belongs in a 'sample characterization' part of the methods (clarifying how sample selection rules worked out)

The parameters DCR and LCR are used to examine the correlation between tree characteristics and root distribution, but are not included in the inclusion sample.

For all 'upslope' versus 'downslope' comparisons as need a much stronger statistical analysis of angular distributions

-- I stopped reviewing here, as you'll need more convincing statistical treatment of your data

We use the angular distribution to perform ANOVA. However, we have a limitation when comparing the roots of each direction. To address this, we perform ANOVA for slope classification and use descriptive analysis (number of roots) to compare upslope and downslope.

Reviewer 4 Report

Comments and Suggestions for Authors

Result

Tree Morphometric

1.      Conversely, the total height of rain trees ranges from 14-25 m, while that of damar trees is 19-25 m. What are the possible reasons? Support your results logically with the help of available literature.

2.      The t-test results showed that tree species significantly affected the average value of tree height, where damar trees had a higher height than rain trees at a significant level of 5%. What are the possible reasons? Support your results logically with the help of available literature.

3.      Notably, the average LCR value of damar trees ((50.41±28.38)%) surpasses that rain trees ((48.28±13.28)%). However, the t-test reveals that tree species had no significant effect on the average LCR value. What are the possible reasons? Support your results logically with the help of available literature.

4.      The t-test results show a significant effect of tree species on the average DCR value. What are the possible reasons? Support your results logically with the help of available literature.

5.      Rain trees have a higher average DCR value ((14.67±2.35)m) incomparison to damar trees ((5.96±1.29)m). Based on slope class, there are no significant differences observed among damar trees. However, in rain trees, class 3 has the lowest value than classes 1, 2, and 4. What are the possible reasons? Support your results logically with the help of available literature.

Soil Physical Properties

6.      The average values of soil properties, including Bulk Density (BD), Porosity (Po), Soil Moisture Content (MCs), and Soil Relative Humidity (RH), for both rain trees and damar trees, regardless of location, are presented in Table 2. What are the possible reasons? Support your results logically with the help of available literature.

7.      The t-test shows that tree type affects the values of bulk density, porosity, and soil moisture content, but not on soil humidity. What are the possible reasons? Support your results logically with the help of available literature.

8.      Descriptively, rain trees have a higher BD value in the up-slope position than down-slope in all classes of land slopes except class 2. What are the possible reasons? Support your results logically with the help of available literature.

9.      In contrast, the down-slope position in damar trees has a higher BD value across all slope classes. What are the possible reasons? Support your results logically with the help of available literature.

10.  The results of the t-test showed that the average porosity value of the soil for damar trees ((70.51±1.64)%) was higher than rain trees ((63.14±3.17)%) at a significant level of 5%. What are the possible reasons? Support your results logically with the help of available literature.

11.  ANOVA test results showed no significant difference between classes and the position of the slope of bulk density of the two species. What are the possible reasons? Support your results logically with the help of available literature.

12.  Meanwhile, the porosity value for rain trees is in a lower up-slope position except for class 2. What are the possible reasons? Support your results logically with the help of available literature.

13.  The results of the t-test showed that the average value of damar trees soil water content ((77.64±6.33)%) was higher than rain trees ((45.79±4.90)%) at a of 5% significance level. What are the possible reasons? Support your results logically with the help of available literature.

14.  Descriptively, the soil water content in the up-slope position is higher than the down-slope for damar trees. However, for rain trees, the soil water content values between slope positions are not different. What are the possible reasons? Support your results logically with the help of available literature.

Root Physics Properties

15.  The results of the t-test demonstrate t tree species significantly impact the value of root moisture content (MCr) (Table 2), where the damar tree had a value of root moisture content ((134.70±21.98)%) higher than rain trees ((89.57±11.25)%). However, tree species did not significantly affect the average root mass density (ρ) value. What are the possible reasons? Support your results logically with the help of available literature.

Root Detection

16.  The results showed that the average sonic velocity at the roots (V roots) was lower than 2000 m.s⁻¹, namely rain trees were (1526.65±183.30) m.s⁻¹ and damar trees were (1709.45±264.77) m.s⁻¹ (Table 2). The t-test results showed that tree species affected the sonic velocity at main roots (V roots) at a significant level of 5%, where the average sonic velocity of damar trees was higher than that of rain trees. However, tree species do not generally affect the average sonic 356 velocity value (V) (Table 2). What are the possible reasons? Support your results logically with the help of available literature.

Sonic Velocity (V)

17.  The correlation test results showed a negative relationship between sonic velocity (V) and soil moisture content (MCs) in damar trees, which was -0.17 (Table 4). However, it had a positive relationship in rain trees but was weak, which was 0.09 (Table 3). Meanwhile, there is a positive relationship between sonic velocity (V) and root moisture content (MCr) in the damar trees, which was 0.19 (Table 4). But in the rain trees, it has a negative relationship of -0.18 (Table 3). This result means that in rain trees with drier soil conditions (MCs = (45.79±4.90)%), sonic velocity is weakly correlated with soil moisture content but negatively correlated with root moisture content. What are the possible reasons? Support your results logically with the help of available literature.

Sonic Velocity of Main Roots (V root)

18.  The results of ANOVA test showed a significant difference in the V root of the slope class on damar trees. The average V root value of class 3 (1531.33 m.s⁻¹) is lower than that of classes 1, 2 and 4. What are the possible reasons? Support your results logically with the help of available literature.

Root Distribution

19.  The root distribution was estimated based on the distribution of the V values of the roots in the up-slope and down-slope positions of the soil. The number of V roots indicates the number of roots detected in the sample tree. The results showed that the average number of roots detected using a root detector was higher than the photogrammetric method (Table 2). The average number of roots detected using a root detector for damar trees (6 ± 2) is higher than rain trees (5 ± 2). Meanwhile, the photogrammetric method showed the opposite value where the number of damar roots (3 ± 1) was lower than rain tree (4 ± 1). However, t-test results showed no effect of tree species on the number of roots in the root detector and photogrammetry methods. What are the possible reasons? Support your results logically with the help of available literature.

Conclusion

1.      Please rewrite conclusion which includes highlights of your results.

2.      What you recommend for future?

3.      What are the benefits of this study?

Comments on the Quality of English Language

Moderate changes required

Author Response

No

Comment

Response

Conversely, the total height of rain trees ranges from 14-25 m, while that of damar trees is 19-25 m. What are the possible reasons? Support your results logically with the help of available literature.

The rain trees has decurrent growth which is caused by weak apical dominance. So th tree high lower than (reference added in the text)

2. The t-test results showed that tree species significantly affected the average value of tree height, where damar trees had a higher height than rain trees at a significant level of 5%. What are the possible reasons? Support your results logically with the help of available literature.

The rain trees has decurrent growth which is caused by weak apical dominance. So th tree high lower than (reference added in the text)

3.Notably, the average LCR value of damar trees ((50.41±28.38)%) surpasses that rain trees ((48.28±13.28)%). However, the t-test reveals that tree species had no significant effect on the average LCR value. What are the possible reasons? Support your results logically with the help of available literature.

Naturaly, damar tree (excurrent) has higher LCR than rain tree (decurrent). However it might be influenced by genetics and physical environment. However, t-test is not sufficient to reject the H0 due to high varience on LCR of damar tree. (references added in the text)

4.The t-test results show a significant effect of tree species on the average DCR value. What are the possible reasons? Support your results logically with the help of available literature.

Naturally, rain tree (decurrent) has higher DCR than damar tree (excurrent) due to a bigger diameter relative to their height.

5.Rain trees have a higher average DCR value ((14.67±2.35)m) incomparison to damar trees ((5.96±1.29)m). Based on slope class, there are no significant differences observed among damar trees. However, in rain trees, class 3 has the lowest value than classes 1, 2, and 4. What are the possible reasons? Support your results logically with the help of available literature.

rain tree (decurrent) has higher DCR than damar tree (excurrent) due to a bigger diameter relative to their height. Sloping land tends to be more easily affected by rainfall, which can cause soil slides so that the fertile topsoil will be washed away (references added in the text). However, in Class 4, the other factor affects the crown tree such as middle slope condition to capture

6.The average values of soil properties, including Bulk Density (BD), Porosity (Po), Soil Moisture Content (MCs), and Soil Relative Humidity (RH), for both rain trees and damar trees, regardless of location, are presented in Table 2. What are the possible reasons? Support your results logically with the help of available literature.

The study found that compaction had the most significant impact on bulk density, porosity, soil moisture, and relative humidity, which in turn affected water and nutrient availability. (references added in the text).

7.      The t-test shows that tree type affects the values of bulk density, porosity, and soil moisture content, but not on soil humidity. What are the possible reasons? Support your results logically with the help of available literature.

We revised this description. A t-test does not establish causality. The difference is caused by compaction, which affects most soil properties. (references added in the text).

8.      Descriptively, rain trees have a higher BD value in the up-slope position than down-slope in all classes of land slopes except class 2. What are the possible reasons? Support your results logically with the help of available literature.

This implies that rainfall removing fine particles from steeper slopes leaves behind a higher concentration of coarser particles, which results in higher bulk densities at these sites. (reference added in the text).

9.      In contrast, the down-slope position in damar trees has a higher BD value across all slope classes. What are the possible reasons? Support your results logically with the help of available literature.

Contrast on results presumably due to soil disturbance due to urban area

10.  The results of the t-test showed that the average porosity value of the soil for damar trees ((70.51±1.64)%) was higher than rain trees ((63.14±3.17)%) at a significant level of 5%. What are the possible reasons? Support your results logically with the help of available literature.

The difference is caused by compaction, which affects to posrosity. (references added in the text).

11.  ANOVA test results showed no significant difference between classes and the position of the slope of bulk density of the two species. What are the possible reasons? Support your results logically with the help of available literature.

The variable that affects bulk density is not sufficient to describe the result. (references added in the text).

12.  Meanwhile, the porosity value for rain trees is in a lower up-slope position except for class 2. What are the possible reasons? Support your results logically with the help of available literature.

The variable that affects bulk density is not sufficient to describe the result. (references added in the text).

13.  The results of the t-test showed that the average value of damar trees soil water content ((77.64±6.33)%) was higher than rain trees ((45.79±4.90)%) at a of 5% significance level. What are the possible reasons? Support your results logically with the help of available literature.

Forested area typically feature soils with higher permeability, greater infiltration, and percolation capacity due to the high level of biological activity (references added in the text).

14.  Descriptively, the soil water content in the up-slope position is higher than the down-slope for damar trees. However, for rain trees, the soil water content values between slope positions are not different. What are the possible reasons? Support your results logically with the help of available literature.

The higher water content in the upslope sue to the porosity of soil. However, it is not significant. Theoritically, up-slope position have lower water content (references added in the text)

15.  The results of the t-test demonstrate t tree species significantly impact the value of root moisture content (MCr) (Table 2), where the damar tree had a value of root moisture content ((134.70±21.98)%) higher than rain trees ((89.57±11.25)%). However, tree species did not significantly affect the average root mass density (ρ) value. What are the possible reasons? Support your results logically with the help of available literature.

Hardwoods (rain tree) have more complicated anatomical features and greater structural variation compared to softwoods (damar tree) , which results a greater range in permeability and capillary behaviour. Wood density did not directly affect the permeability (references added in the text)

16.  The results showed that the average sonic velocity at the roots (V roots) was lower than 2000 m.s⁻¹, namely rain trees were (1526.65±183.30) m.s⁻¹ and damar trees were (1709.45±264.77) m.s⁻¹ (Table 2). The t-test results showed that tree species affected the sonic velocity at main roots (V roots) at a significant level of 5%, where the average sonic velocity of damar trees was higher than that of rain trees. However, tree species do not generally affect the average sonic 356 velocity value (V) (Table 2). What are the possible reasons? Support your results logically with the help of available literature.

Damar tree has higher MCs and MCr than rain tree, high water content in wood or soil tends to slow down the speed of wave propagation. It suggested that velocity more affected by anatomical features (references added in the text)

17.  The correlation test results showed a negative relationship between sonic velocity (V) and soil moisture content (MCs) in damar trees, which was -0.17 (Table 4). However, it had a positive relationship in rain trees but was weak, which was 0.09 (Table 3). Meanwhile, there is a positive relationship between sonic velocity (V) and root moisture content (MCr) in the damar trees, which was 0.19 (Table 4). But in the rain trees, it has a negative relationship of -0.18 (Table 3). This result means that in rain trees with drier soil conditions (MCs = (45.79±4.90)%), sonic velocity is weakly correlated with soil moisture content but negatively correlated with root moisture content. What are the possible reasons? Support your results logically with the help of available literature.

High water content in wood or soil tends to slow down the speed of wave propagation, that why most of MCs and MCr are negative. Event some values are positive value, but it is not significant. I can be caused by anatomical feature and distance between root and tool. (references added in the text)

18.  The results of ANOVA test showed a significant difference in the V root of the slope class on damar trees. The average V root value of class 3 (1531.33 m.s⁻¹) is lower than that of classes 1, 2 and 4. What are the possible reasons? Support your results logically with the help of available literature.

The ANOVA analysis was changed to an angular distribution analysis based on the reviewer's suggestion. The results indicated no significant differences.

19.  The root distribution was estimated based on the distribution of the V values of the roots in the up-slope and down-slope positions of the soil. The number of V roots indicates the number of roots detected in the sample tree. The results showed that the average number of roots detected using a root detector was higher than the photogrammetric method (Table 2). The average number of roots detected using a root detector for damar trees (6 ± 2) is higher than rain trees (5 ± 2). Meanwhile, the photogrammetric method showed the opposite value where the number of damar roots (3 ± 1) was lower than rain tree (4 ± 1). However, t-test results showed no effect of tree species on the number of roots in the root detector and photogrammetry methods. What are the possible reasons? Support your results logically with the help of available literature.

The photogrammetry method only captures the shallow roots that are visible on the surface, while root detectors can detect both above and underground roots. The results of the photogrammetry and root detector tests showed that the rain tree roots were shallower, as they could be visually detected, while the damar tree roots were deeper.

Please rewrite conclusion which includes highlights of your results.

Conclusion has been revised

What you recommend for future?

Based on PCA analysis, we can reduce the close variable due to limited sample

 What are the benefits of this study?

The root detector is a non-invasive technology that can identify the main root of a tree without causing damage. It is also relatively cheaper than GPR. This study established method for mature/large tree in some environment variation

Round 2

Reviewer 2 Report

Comments and Suggestions for Authors

Dear colleagues!

The article “Clarifying Main Root Distribution of Trees in Varied Slope Environments Using Non-Destructive Root Detection” has been significantly improved by the authors. Successful correction of the main sections of the article was carried out. Although the possibility of applying PCA to the received data is still questionable (the sample is insufficient). It is advisable to describe the limits of PCA applicability in the Discussion section.

Reviewer's conclusion: reconsider after minor revision.

Comments on the Quality of English Language

Dear colleagues!

The article “Clarifying Main Root Distribution of Trees in Varied Slope Environments Using Non-Destructive Root Detection” has been significantly improved by the authors. Successful correction of the main sections of the article was carried out. Although the possibility of applying PCA to the received data is still questionable (the sample is insufficient). It is advisable to describe the limits of PCA applicability in the Discussion section.

Reviewer's conclusion: reconsider after minor revision.

Author Response

Thank you for your critical feedback and suggestions that have helped enhance the scientific value of our work, making it a better piece of writing. We have considered all the comments and tried to incorporate the suggestions.

No

Comment

Response

Although the possibility of applying PCA to the received data is still questionable (the sample is insufficient). It is advisable to describe the limits of PCA applicability in the Discussion section.

PCA was used to evaluate the main variables due to the same proxies. It can reduce variables in future research.

 We have reviewed the terms of use of the PCA. Even though there is a limited sample size, the research design still meets the statistical analysis requirements according to Akuoma et al. (2020), which states that if the number of samples (n) is the same as the number of variables (p) then PC can be used. Meanwhile, in the research, we conducted, n > p, so we were still eligible to use PC, i.e., n= 24 and p= 11

Akuoma Mabel, O.; Samuel Olayemi, O. A Comparison of Principal Component Analysis, Maximum Likelihood and the Principal Axis in Factor Analysis. American Journal of Mathematics and Statistics 2020, 10(2): 44-54, doi: 10.5923/j.ajms.20201002.03.

Reviewer 3 Report

Comments and Suggestions for Authors

Please check that response to reviewers matches actual changes to the manuscript

Author Response

Thank you for your critical review. We have carefully paid attention to every comment, suggestion, and input from reviewers and made revisions as best as possible. We have included several reference suggestions, improvements to the analysis and additional explanations have also been made. So, we think the quality of the manuscript is getting better. We have attached previous comments and responses to the manuscript.

Thank You

No

Comment

Response

The test of the 'sonic root detector' under other circumstances than for which it has been developed is welcome. Questions of root orientation on slopes are relevant -- although there are alternative hypotheses on functional aspects (tree mechanical stability, water & nutrient uptake opportunities) that are only hinted at here, rather than analyzed.

The excurrent-decurrent contrast that's in the title would require a substantially larger data set (more tree species) for testing rather than 1 species in each class. Please leave it out of title and abstract; other differences, such as wood density, offer more direct explanations for your results.

Tree mechanical properties were represented by LCR and DCR, while environment parameters were captured by RH, Rho, dan MC. All those parameters were analyzed based data that we have.

Thank you for your suggestion. We have taken it into consideration and updated our title accordingly.

We removed crown shape (excurrent and decurrent) in the title. However, in the abstract, we are still considering differentiating the crown shape due to load and root distribution; wood density between species is not very different.

The method descriptions that you refer to suggest the receptor sensors to be placed at 6 x DBH, rather than a standardized 80 cm; you'll need to justify this deviation

Based on our previous study on mature trees and the reference study conducted by Proto et al, it has been determined that when using small plants, the maximum distance for consistent velocity is 200cm. However, at 120cm, the velocity is not consistent. After analyzing both our study and the reference, it has been concluded that the optimum distance for this method is 80cm.

We can be explained that:
“ 6 x DBH as suggested in Proto et al  (2020) is not possible in this study which using big diameter tree, so we use 80 cm as the best distance as applied by Proto et al (2020) and Rahman et al. (2023)”

Information on DBH is missing -- yet, it has been used as a scaling parameter for many tree properties, including proximal root data -- as in a study on Indonesian trees on sloping land:

Hairiah, K., Widianto, W., Suprayogo, D. and Van Noordwijk, M., 2020. Tree roots anchoring and binding soil: Reducing landslide risk in Indonesian agroforestry. Land, 9(8), p.256.

Tree stability on DCR has been established by using DBH as a proxy. Additionally, the reference has been added for further clarification.

We added the reference however we accommodated the reference for introduction due to our study is not focus on DBH on root anchorage; we more focus on crown (DCR and LCR). The reason for using DCR in this study was its association with root distribution

Title could be: Coarse root distribution of trees on slopes clarified by a sonic root detector

Based on your suggestion, we have changed the title.

Introduction: The first two sentences can be omitted, as they hardly prepare for what follows.

Introduction might position the 'sonic root detector' in a tradition of studying electrical conductance (Chloupek, O., 1977. Evaluation of the size of a plant's root system using its electrical capacitance. Plant and Soil, 48(2), pp.525-532. 

Dalton, F.N., 1995. In-situ root extent measurements by electrical capacitance methods. Plant and soil, 173, pp.157-165.  

Dietrich, R.C., Bengough, A.G., Jones, H.G. and White, P.J., 2013. Can root electrical capacitance be used to predict root mass in soil?. Annals of Botany, 112(2), pp.457-464.)

Ellis, T.W., Murray, W., Paul, K., Kavalieris, L., Brophy, J., Williams, C. and Maass, M., 2013. Electrical capacitance as a rapid and non-invasive indicator of root length. Tree physiology, 33(1), pp.3-17.)

and ground-penetrating radar (Hruska, J., Čermák, J. and Šustek, S., 1999. Mapping tree root systems with ground-penetrating radar. Tree physiology, 19(2), pp.125-130.

Guo, L., Chen, J., Cui, X., Fan, B. and Lin, H., 2013. Application of ground penetrating radar for coarse root detection and quantification: a review. Plant and soil, 362, pp.1-23.

Butnor, J.R., Doolittle, J.A., Johnsen, K.H., Samuelson, L., Stokes, T. and Kress, L., 2003. Utility of ground‐penetrating radar as a root biomass survey tool in forest systems. Soil Science Society of America Journal, 67(5), pp.1607-1615.) summarizing strengths and weaknesses of both methods. The sonic root detector appears to be easier and lower-cost, with less sensitivity to soil water content (and no sensitivity to electrolyte contents that hamper interpretation of the electical conductance   data)

Thank you for the suggested references. We have considered and cited them. Additionally, we have revised the introduction.

Line 89 'choppy'? please check classification system used; more relevant is the actual slope angle on which the trees grew

In the reference, there is no detail about the classification. We rephrased the sentence to describe that the location has varying slopes and elevations.

Line 140 How were the destructive root samples collected?

As per the methodology, we obtained small root samples, each measuring 2x2x1 cm, from two primary roots - one from the up-slope and the other from the downslope. In order to determine the root density, two samples were collected from each tree, and the Archimedes method was used.

Section 2.5: more detail is needed on how the method was used and how data are interpreted (even if the details may be part of proprietary software): please outline the principles used 

Section 2.5 now provides a more comprehensive explanation of the tool's working principle.

Figure 3C: the dotted lines connecting tree center to local maxima in sonic velocity are probably not the most likely directions of roots... Have alternative methods of interpretation been tested?

To uncover the coarse roots, we rely on visual methods, specifically photogrammetry, as illustrated in Figure 3b. Proto's 2020 study tested root detection through excavation, but it was found that the direction of detected roots could provide proximity to the actual root direction, rather than being exact.

Line 202: what does this mean? please reformulate

The sentence regarding the range of slopes within the same classification has been revised to be more clear.

Section 2.7 Beyond the software used, the reader will want to know what level of 'independence' and 'replication' was tested, as the confounding of species x location could not be unpacked. How did you construct 'random' null hypotheses to be rejected (or not)? The angular distribution has some specific challenges. ANOVA application needs more detail.

We utilized angular distribution in our statistical analysis and the ANOVA.

The root velocity observations in this study are not independent of each other, so we transform them into an angular distribution.

We do not compare trees because they have different characteristics, but we differentiate them based on their location.

Then we perform anova for each parameter.

3. Results and Discussion: please separate into two sections, and first give the results, before discussing your questions

We separate it into two sections

Line 218: this is not a result, but a direct part of your sample selection. You need to clarify the criteria you used for inclusion in the sample.

We moved that part in the method

Line 219: given this tee diameter, your deviation from the 6xDBH advise is very considerable...

We agree that the best distance to identify root biomass is at 6xDBH. However, the root detector cannot detect roots beyond 200 cm from the trunk center. Proto et al established that the 6xdbh threshold applies only to trees up to 120, rendering it inapplicable to larger trees.

Line 221 "The t-test results showed that tree species significantly affected the average value of tree height" NO! the t-test shows that it is unlikely that the two tree species belong to a homogeneous distribution, no causality implied 

We have made changes to the sentences and conducted a t-test to confirm differences in tree species characteristics.

Fig 5 This belongs in a 'sample characterization' part of the methods (clarifying how sample selection rules worked out)

The parameters DCR and LCR are used to examine the correlation between tree characteristics and root distribution, but are not included in the inclusion sample.

For all 'upslope' versus 'downslope' comparisons as need a much stronger statistical analysis of angular distributions

-- I stopped reviewing here, as you'll need more convincing statistical treatment of your data

We use the angular distribution to perform ANOVA. However, we have a limitation when comparing the roots of each direction. To address this, we perform ANOVA for slope classification and use descriptive analysis (number of roots) to compare upslope and downslope.
